# Neural and behavioral signatures of the multidimensionality of manipulable object processing

Jorge Almeida [1,2✉], Alessio Fracasso[3], Stephanie Kristensen[1,2], Daniela Valério[1,2], Fredrik Bergström[1,2,4], Ramakrishna Chakravarthi[5], Zohar Tal [1,2] & Jonathan Walbrin[1,2]

Understanding how we recognize objects requires unravelling the variables that govern the way we think about objects and the neural organization of object representations. A tenable hypothesis is that the organization of object knowledge follows key object-related dimensions. Here, we explored, behaviorally and neurally, the multidimensionality of object processing. We focused on within-domain object information as a proxy for the decisions we typically engage in our daily lives – e.g., identifying a hammer in the context of other tools. We extracted object-related dimensions from subjective human judgments on a set of manipulable objects. We show that the extracted dimensions are cognitively interpretable and relevant – i.e., participants are able to consistently label them, and these dimensions can guide object categorization; and are important for the neural organization of knowledge – i.e., they predict neural signals elicited by manipulable objects. This shows that multidimensionality is a hallmark of the organization of manipulable object knowledge.

[1] Proaction Lab, Faculty of Psychology and Educational Sciences, University of Coimbra, Coimbra, Portugal. [2] CINEICC, Faculty of Psychology and Educational Sciences, University of Coimbra, Coimbra, Portugal. [3] School of Psychology and Neuroscience, University of Glasgow, Glasgow, UK. [4] Department of Psychology, University of Gothenburg, Gothenburg, Sweden. [5] School of Psychology, University of Aberdeen, Aberdeen, UK. ✉email: jorgecbalmeida@gmail.com

Our ability to recognize one object amongst many others is one of the most important features of the human mind, and successful completion of our daily activities depends upon it. This is nicely illustrated by the performance of neurological patients that present with semantic impairments, and are severely dysfunctional even when performing mundane activities[1–5]. Our capacity to recognize objects is most probably dependent on how we represent object-knowledge and on how these representations are organized in the brain[6–8]. One tenable hypothesis on the organization of object knowledge holds that object representations are, at some level, grounded on, and framed by, relevant object-related dimensions in which objects vary[9–13]. These object-related dimensions would allow for the hard, fine-grained, and most times within-category distinctions that are typically required when recognizing objects. However, there are substantial gaps to be closed under such a hypothesis – most importantly, what are these object-related dimensions that govern the fine-grained neural organization of object representations? Here, we will extract object-related dimensions from participants' understanding of objects, and show that these dimensions are cognitively and neurally relevant.

Research on the organization of object knowledge in both human and non-human primates points to the existence of clusters of neurons that show categorical preferences for particular domains such as faces, animals, body parts, places, and manipulable objects[12,14–23]. Moreover, data from brain-damaged patients also points to the presence of cognitive modules dedicated to those same domains[1–4,24]. In ventral temporal cortex, these object preferences follow a lateral-to-medial organization with clusters of neurons preferring faces or animals situated within lateral fusiform gyrus, and clusters of neurons preferring places or manipulable objects located more medially[8,25]. This has been described either as 1) a sharp functional division reflecting true domain distinctions[1,3,15,26–31], laid out by connectivity constraints between these object-preferring regions and other regions elsewhere in brain that code for the same domain[32–34]; or as 2) a continuous map of similarity over one single dimension such as animacy[35], or shape[36,37] (for other dimensions see refs. [38,39]).

Thus, most current approaches propose overarching explanations for the general organization of object knowledge by object-preferring regions, in lieu of focusing on the fine-grained organization of conceptual content within these different regions. That is, their focus has been on explaining the pattern of results from neuropsychology and neuroimaging at a general level, appealing to a first principle of organization that has to be somehow related with between-domain differences[1,3,27–39]. These efforts, while necessary, may have deterred the field from also explaining the finer-grained distinctions – those that seem to be affected in category-specific brain-damaged patients[1–4]. Understanding how we identify a hammer from other objects such as axes, flyswatters, or screwdrivers – i.e., those kinds of distinctions that we probably have to deal with every day, and that are affected in category-specific deficits – requires uncovering finer-grained types of information than identifying a hammer from a cat, a truck, or a mimosa.

To solve this, we may think of object representations as being organized in particular multidimensional spaces. These spaces preserve individual properties of objects, while reducing the complexity that is inherent to object recognition, by situating each object's representation within key object-related dimensions. Recently, there have been some attempts at describing the multidimensional space that underlies object knowledge[13,24,40–42]. For instance, Hebart and collaborators have suggested a series of dimensions explain, at a general level, how individuals represent many different objects[13]. The dimensions obtained are relatively interpretable (e.g., colorful; fire-related) and are able to predict performance in an odd-one-out similarity judgement task. While these efforts certainly advance our understanding of the processes at play during object recognition, they still follow a "between-category" approach. Understanding fine-grained decisions is probably better served (or at least it is also served) by studies that focus on "within-category" strategies.

In this paper, we focus on demonstrating that a series of object-related dimensions extracted from participants' understanding of manipulable objects, relate to the neural representations of those objects, and guide object perception. We selected manipulable objects because they have a set of properties that are useful when trying to define object-related dimensions. Namely, 1) they are everyday manmade objects that we perceive and interact with constantly, and are, thus, fairly familiar; 2) they hold relatively defined sets of information associated with them: by definition these objects have particular functions that they fulfill, have associated motor programs for their use, and have specific structural features (e.g., shape) that may help fulfill both their function and facilitate their manipulation; and 3) their visual inspection engages a set of neural regions that includes aspects of the left inferior parietal lobule (IPL), the anterior intraparietal sulcus (aIPS), bilateral superior and posterior parietal cortex (SPL) and caudal IPS, bilateral dorsal occipital regions proximal to V3A, the left posterior middle temporal gyrus (pMTG), and bilateral medial fusiform gyrus[14,18–21,26,43–49]. Moreover, conceptual knowledge about manipulable objects can be selectively impaired or spared in brain damaged patients[2].

Here, we will independently capture subjective object similarity in terms of the visual properties of the target objects, the manner with which we manipulate these objects, and the function that is typically associated with them. We selected these knowledge types (i.e., vision, function and manipulation) because these are central for the representation of manipulable objects. We hypothesize that: 1) participants can learn to recognize and categorize objects according to these dimensions, and 2): sensitivity to these dimensions will be captured with neural responses.

Specifically, we expect that the dimensions extracted conform to general major subdivisions within each knowledge type, and that the spatial extent of the neural responses explained by each dimension reflects the type of content they represent. For instance, dimensions that structure our understanding of object function should revolve around the action goals that we fulfill with manipulable objects (e.g., cleaning, cooking, cutting, writing), and/or the context where an object is typically encountered in (e.g., kitchen; bathroom;[32,50–52]). As such, these dimensions should explain neural responses elicited by manipulable objects within pMTG and lateral occipital cortex (LOC), ventral temporal cortex in the vicinity of parahippocampal gyrus, as well as potentially more anterior temporal regions[32,50–52]. On the one hand, pMTG and LOC have been shown to code for object-related action knowledge and meaning[53–58]. That is, these regions seem to be involved in understanding the functional goals of the actions we can perform with objects. On the other hand, medial and anterior aspects of ventral temporal cortex are involved in processing of spatial relations, and different (visual) environments[22,30] – and thus relate to the context in which an object is encountered.

Dimensions that structure our understanding of the manner in which an object is manipulated should relate both to motor aspects that are directly available from the visual input (e.g., object affordances such as grasp types; e.g., see refs. [59–67]), as well as to aspects that may have to be derived from the visual input and are part of an object-specific manipulation program (e.g., object-related specific movements such as different wrist rotations). These dimensions should be able to explain responses in regions that relate to the processing of affordances and praxis.

These include occipito-parietal and posterior and superior parietal cortical regions (in the vicinity of V3A, V7, and IPS) that have been shown to be important for the computation of object grasps and object affordances[59–67], and for 3D object processing[68–78]. Moreover, the left IPL, which has been shown to be causally involved in processing object-specific praxis[14,51,79,80], should also be explained by dimensions structuring manipulation similarity.

Finally, we expect that the way in which we understand the visual properties of an object to be organized under two major types of object-related dimensions: those that relate to geometric properties (e.g., shape, size, elongation); and those that relate to surface and material properties (e.g., type of material, shininess, color). This distinction has been shown consistently in several single case patient studies. For instance, Patient DF presented with a clear deficit in processing visual form in the context of spared processing of surface and material properties[81–83], whereas patient MS presented with deficits that are specific to the processing of surface properties, in the context of spared shape information[82,83]. This distinction is further supported by neuroimaging studies that demonstrate that more medial aspects of ventral temporal cortex (from lingual gyrus anteriorly to the parahippocampal gyrus) participate in the computation of surface and material properties of the visual stimuli[82–85], whereas more lateral and posterior aspects of ventral temporal cortex[82–84,86–88], and dorsal occipital cortex[19,69,89–91] code for geometric properties. Thus, we expect the dimensions we obtain that structure visual similarity to relate, independently, to geometric and to surface properties and materials, and to conform to these lateral-to-medial, posterior-to-anterior neural dissociations.

Overall, we predict that these dimensions figure critically in how we represent manipulable objects, and potentially support within-category distinctions. Focusing on this level of analysis will complement the literature on object recognition that has typically addressed conceptual knowledge at a between-domain level of analysis.

## Results

**Overview**. We first selected a set of 80 common manipulable objects (see Table S1 for all the objects; see Methods). These were selected to be representative of the different types of objects used routinely. We then obtained similarity spaces for these 80 manipulable objects in the 3 knowledge types ($N = 60$), and extracted key object-related dimensions that govern our manipulable object representational space using non-metric multidimensional scaling (MDS[92,93]; Fig. 1A; see Methods). The selected dimensions were subsequently labelled by a second group of participants ($N = 43$) in order to ascertain their interpretability (Fig. 1B). We then tested whether these dimensions could guide participants' behaviors. We taught participants ($N = 210$) to categorize a subset of our 80 objects in terms of their scores along a target dimension, and tested whether their learning could be generalized to a subset of untrained items. We considered an untrained categorization task. However, determining which dimension participants would use (or whether they would use one single dimension consistently throughout the experiment) would likely result in large differences in interpretation of the task across subjects. Instead, we used a learning paradigm to specifically test whether participants could reliably learn the organization of each dimension. Finally, we tested whether the extracted dimensions explained neural responses to manipulable objects. We developed an event-related functional magnetic resonance imaging experiment (fMRI; see Fig. 1C) where we presented a new group of participants ($N = 26$) with images of the 80 manipulable objects (see Figure S1 for examples of the images used in this experiment). We used parametric analysis[94–96], casting our key dimensions as first-order (i.e., linear) parametric modulators in a General Linear Model (GLM; see Fig. 1C), and asked whether the scores of each object in each dimension were able to explain neural responses elicited by those same objects. Using parametric modulations is the most appropriate approach because of the continuous nature of the scores of the objects in the dimensions extracted with MDS. Under this approach, we can directly test (i.e., without transforming the data) whether the responses of a voxel are a function of the scores in the target dimension.

**Extraction and labelling of key object-related dimensions**. The number of dimensions in the final MDS solution per knowledge type was determined based on *stress* value (Kruskal's normalized stress[93]). The final solutions led to the extraction of 4 orthogonal functional dimensions (stress value for the solution 0.09), 6 orthogonal manipulation dimensions (stress value for the solution 0.08), and 5 orthogonal visual dimensions (stress value for the solution 0.09; see Fig. 2 for these dimensions; see Figure S2 for scree plots; see Methods for details).

An initial test of our dimensionality solution is to understand how interpretable these dimensions. We collected and collated labels for each of the extracted dimensions. In Fig. 2, we present a rank order distribution of the 20 most extreme objects per dimension (10 from each extreme), along with the selected label and the collated percent label generation score (a visualization of the labels generated can be found in Fig. S3). For most dimensions, participants reliably reached similar interpretations. The most consensual labels generated for the visual dimensions include aspects related with shape (e.g., roundedness), size, or material properties (e.g., presence of metal); the labels for the functional dimensions include aspects related to different types of activities performed (e.g., cleaning vs construction), or the context in which the objects are seen (e.g., kitchen vs. office); and labels for the manipulation dimensions include aspects related to grasp types (e.g., power vs. precision), different types of motion (e.g., rotation), and object properties that relate downstream to object manipulation (e.g., need for force or dexterity). There are some dimensions whose labels were not consensual (e.g., dimension 5 of vision present label generation percent scores of about 20%). Interestingly, these lower scores are observed more consistently in later (e.g., the fifth dimension) rather than earlier (e.g., the first dimension) dimensions, as this may relate to the fact that these dimensions explain less of the total variance. Overall, however, the reliability of the labels attributed to each dimension seems a first pass demonstration of the feasibility of our approach.

**Object-related dimensions guide object categorization behavior**. A much stronger test of the centrality of these dimensions is to understand whether they can guide behavior. Figure 3 shows categorization performance on untrained manipulable objects, as to whether the target objects were more categorizable as closer to one or the other extreme object in a target dimension. To analyze these data, we first averaged responses for untrained objects per bin according to the score of the objects in the target dimension (i.e., 10 bins of 8 consecutive sets of items across the length of each dimension; see Methods). We then fitted a cumulative Gaussian curve to the bin-specific percent categorization responses towards the object with the highest score in the target dimension. We expected that if the object-dimensions were cognitively important for our ability to process and recognize objects, percent responses towards the extreme object with the highest score should increase as a function of the increase in the dimensional score per bin. R-square values per participant

## A. Similarity measures and object-related dimensions

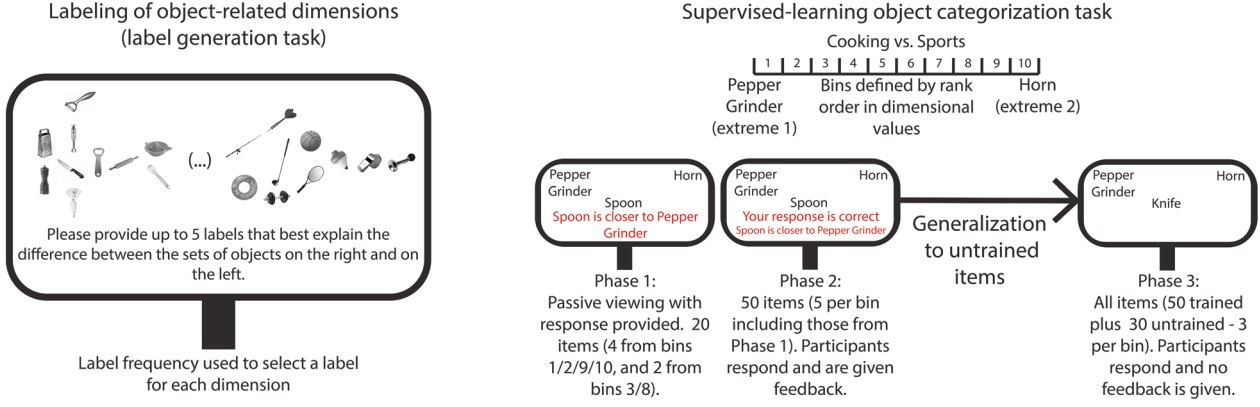

## B. Testing cognitive interpretability and cognitive importance of the object-related dimensions

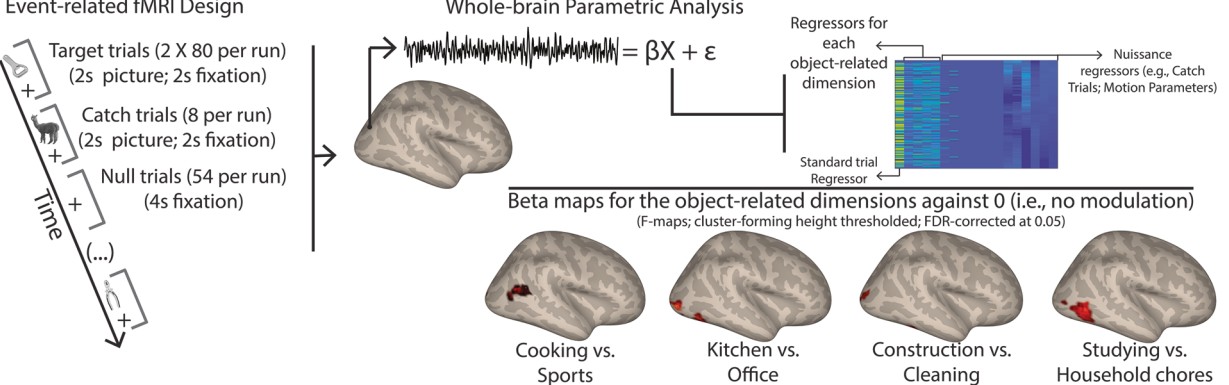

## C. Neural signatures of object-related dimensions

were obtained as a measure of goodness of fit between the cumulative Gaussian curve and each participant's data.

Crucially, participants were able to generalize their learning on trained objects to untrained objects –the extracted dimensions have high explanatory power, indexed by high R-square values (violin plots in Fig. 3E). Moreover, there was a trend for lower goodness of fit for those dimensions that were lower in the stress ranking of our MDS solution (the last dimensions of function and vision and the second to last dimension of manipulation). Furthermore, it is also very clear that the control conditions (i.e., lexical frequency and a random item ordering; see Methods) did not explain the behavioral data, as indicated by their lack of learning generalization (and much weaker R-square values when compared to those obtained for the key object-related dimensions).

**Object related-dimensions predict neural responses elicited by manipulable objects**. A final stage in demonstrating that multi-dimensionality is a signature of high-level object processing, and specifically of manipulable object processing, is to demonstrate that the organization of our neural representations about manipulable objects also relies on this multidimensional space.

As can be seen in Fig. 4, our object-related dimensions can account for neural signal elicited by viewing manipulable objects – all key object-related dimensions show linear slope (beta) values that significantly differ from zero. That is, all dimensions significantly modulate the neural responses to manipulable objects in different regions as a function of the object dimensional values. Note that the function dimension "Cooking Vs. Sports" shows limited coverage under a cluster-forming height-thresh-olding of $p < 0.001$, and the dimension "Material properties" fails

**Fig. 1 Experimental procedures and analysis pipeline. A** In order to extract object-related dimensions we collected similarity measures between our 80 manipulable objects through a pile-sorting experiment[92,93]. Per individual ($N = 60$ particpants), we obtained a piling solution for each of the knowledge types (function, manipulation and vision) whereby objects piled together were similar to one another but different from objects in other piles. These piling solutions were coded into dichotomous matrices that represented pile membership. Participant-specific matrices were then averaged and transformed into a dissimilarity matrix (one per knowledge type). Finally, we used non-metric MDS[92,93] to extract dimensions independently per knowledge type. **B** We wanted to test whether the obtained object-related dimensions were cognitively important for perceiving objects. Firstly, we had a different set of participants perform a label generation task for each dimension. Participants were presented with 20 objects – 10 from each of the extremes of the target dimension – and were asked to provide up to 5 labels that best explained the difference between the two sets of objects. Label frequency was used to select a label for each of the extracted dimensions. We further tested the importance of these object-related dimensions by having yet another set of participants learn to categorize objects according to their scores in each of the dimensions. Participants went first through 2 experimental phases where they were taught to categorize a subset of the objects in terms of whether they were close to one of the two extremes of a target dimension and were given clear feedback as to the correct responses. Importantly, in a third phase, they were asked to categorize all objects, including a subset of untrained objects, and were not given any feedback. Moreover, we added two control dimensions. In one of these controls, we took one of the real dimensions and randomly shuffled the scores of the dimension for the individual objects. For a more stringent control, we took lexical frequency values[112] – i.e., count of the times a particular lexical entry appears in a text corpus per million – for each of the objects and rank ordered them in terms of these values. We used these dimensions to control for reliable generalization of object-related dimensional learning to untrained items. We tested whether participants generalized their learning to untrained items. Percent response performance towards the extreme object with the highest score in the dimension was calculated and fitted with a cumulative Gaussian curve. **C** Finally, we tested whether the object-related dimensions extracted were able to explain neural responses to objects. We presented the 80 objects in an event-related fMRI experiment using greyscale images, and participants had to categorize each image as either a manipulable object or an animal (the catch trials). We then used parametric mapping to analyze the fMRI data, and tested whether our object-related dimensions could explain the neural responses elicited by the 80 manipulable objects. We used parametric analysis[94-96] over the fMRI data, and cast our key dimensions as first-order (i.e., linear) parametric modulators in a General Linear Model (GLM). That is, for each stimulus in the design matrix, the corresponding dimensional scores were assigned as modulation values.

to survive correction at that level. Nevertheless, these two dimensions show significant results under a slightly more lenient cluster-forming height-thresholding of $p < 0.005$ (for individual maps of all dimensions see Fig. S4; $N = 26$ participants).

Moreover, these object-related dimensions appear to capture signal variability in or around similar regions, in part within those regions that show a preference for manipulable objects[14,18–21,26,34,43–49]. This is in line with the role of dimensionality in explaining neural organizing of information: perhaps in the same way as the different dimensions that rule the organization of low-level sensory-motor cortices overlap spatially, so do dimensions that rule the organization of manipulable object knowledge in the brain.

**Object related-dimensions maintain content specificity in their neural responses.** Interestingly, and as predicted, these results show specificity as a function of the type of content these dimensions represent. This can be seen more generally when we observe the neural parametric maps of these dimensions grouped by knowledge type. For instance, as a group, function dimensions seem to collectively explain responses (i.e., their responses are in proximity of each other) in or around pMTG and lateral occipital cortex (LOC; see Fig. 4). Manipulation dimensions show more proximity in their explanatory power within occipito-parietal regions (in the vicinity of V3A, caudal IPS, and Precuneus), and also within medial ventral temporal cortical regions, posterior lingual gyrus and pMTG/LOC. Finally, vision dimensions collectively explain responses in ventral temporal cortex and posterior lingual gyrus, as well as in lateral temporal cortex, and in more posterior dorsal occipital regions.

Specificity in the multidimensional organization can also be observed when we look at these dimensions individually (individual parametric F-maps against 0 – i.e., no modulation – can be seen in Fig. S4; The F-maps for the first two dimensions of each knowledge type are presented in Fig. 5). These maps show particular aspects that relate to the content represented by each of these dimensions.

The first two function dimensions ("Cooking vs. Sports" and "Kitchen vs. Office") explain neural responses within broader pMTG/LOC, although the dimension that relates to the physical context in which the objects are typically observed ("Kitchen vs. Office") explained activations in more inferior aspects, whereas the dimension that relates to particular types of activity ("Cooking vs. Sports") explained more superior activation (see Fig. 5). Moreover, the dimension "Cooking vs. Sports" significantly explains activation within right lingual gyrus extending superiorly to the cuneus. Finally, the dimension "Kitchen vs. Office" explains signal also within the border between left IPL and post-central gyrus, and, importantly, within the parahippocampal gyrus.

The two manipulation dimensions also lead to different content-specific modulatory maps. The "Power vs. Precision" dimension is capable of explaining activity within regions of the posterior parietal cortex – namely aIPS (extending to post-central gyrus), and regions proximal to V3A. The dimension "Force vs Dexterity", shows no occipito-parietal and posterior parietal activations, but shows bilateral ventral (both posterior within lingual gyrus, and anterior within parahippocampal gyrus) and lateral occipito-temporal activation (see Fig. 5).

Finally, the two selected visual dimensions also show differences in the spatial extent of their explanatory power. The largest clusters of activation for the dimension "Metal vs. other materials" are medially along the collateral sulcus (posterior to anterior) bilaterally, parahippocampal gyrus and the posterior lingual gyrus. The dimension "Elongated vs. Round" shows a major cluster of explained signal in lateral ventral temporal cortex, as well as in inferior occipital cortex, and also strongly in some aspects of dorsal occipital cortex (Fig. 5).

Overall, our results show that neural responses elicited by manipulable objects are explainable by their scores on key object-related dimensions particularly within regions that typically prefer manipulable objects to other categories of objects (e.g., faces). Moreover, the ability of these dimensions to explain neural responses to objects seems to be related with the kinds content that the dimensions represent.

## Discussion

Unravelling the organization of object knowledge in the brain is a necessary step in understanding how we recognize objects[6–8]. Importantly, most efforts to understand object recognition and

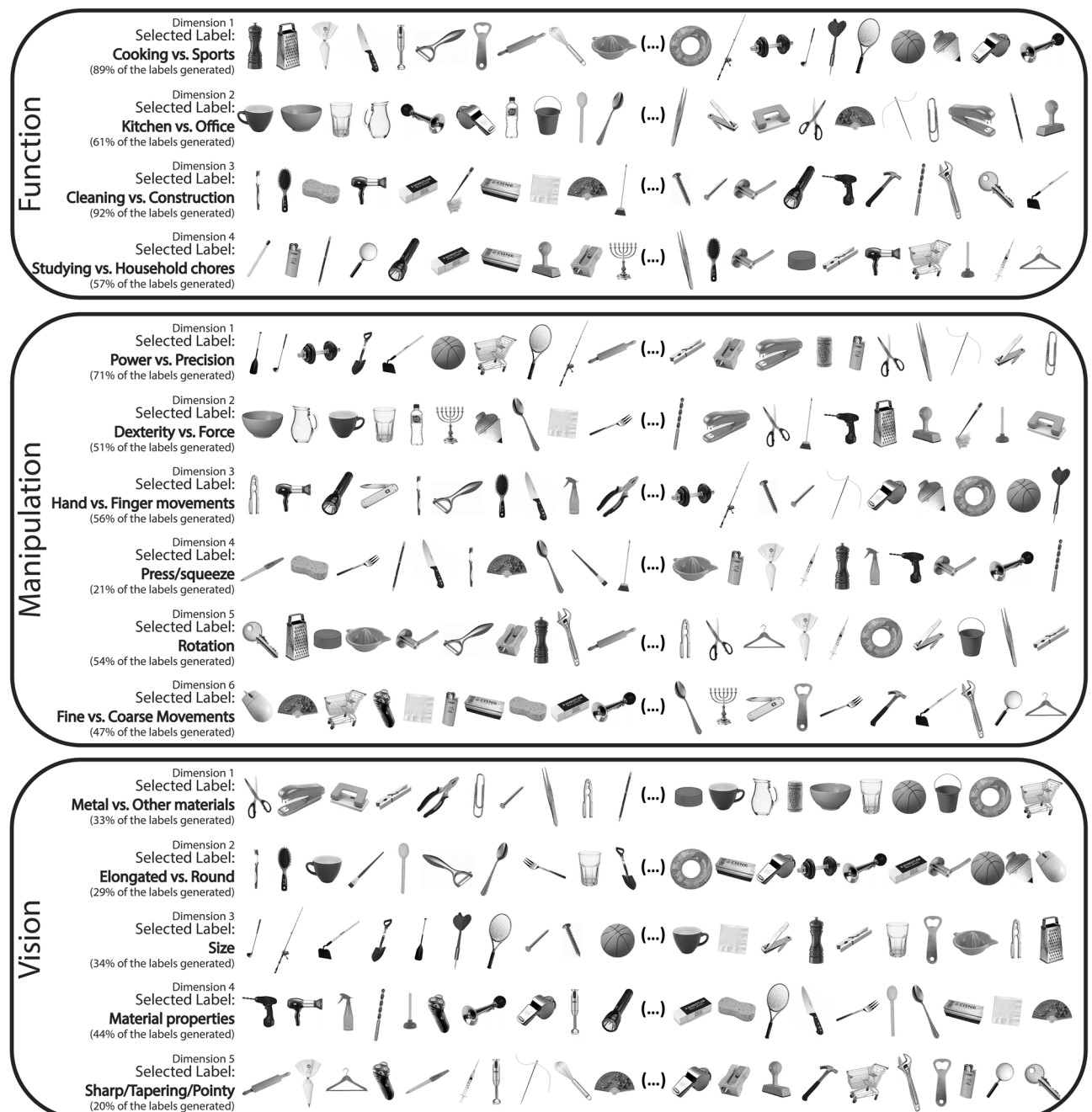

**Fig. 2 Interpretability of the objected-related dimensions.** Here, we present the selected 15 dimensions that govern the internal representation of our set of manipulable objects. Per dimension, we present the 20 objects with the most extreme scores (10 from each extreme). Labels were then selected based on the frequency of label generation by the participants. On the left side of each dimension, we present the collated selected label, as well as the percent generation score for that label (see Figure S3 for the label frequency plots).

the organization and representation of conceptual information in the brain, have focused on explaining between-category differences, or on finding an overarching explanation for object representation. However, the available neuroimaging and neuropsychological data[1,3,4,12,16–18,20–23,29,97–99] seem to point to the parallel need to also look into finer-grained distinctions – i.e., into within-category organization of information. Here, we tested a principled way to explore the multidimensionality of object processing focusing on the category of manipulable objects. Specifically, we extracted object-related dimensions from human subjective judgments on a large set of manipulable objects. We did so over several knowledge types (i.e., vision, function and

manipulation), whose selection was motivated by the typical characteristics that best describe these objects. Our results demonstrate that these dimensions are cognitively interpretable, that they can guide our ability to categorize and think about objects, and that they explain neural responses elicited by the mere visual presentation of these objects. Moreover, our results show that these dimensions are highly generalizable across individuals and modalities of presentation of the stimuli (words vs. pictures), suggesting some level of universality.

Our data seems to conform to some of the conceptual and neural patterns we predicted for the different knowledge types. As expected, dimensions pertaining to function knowledge revolved

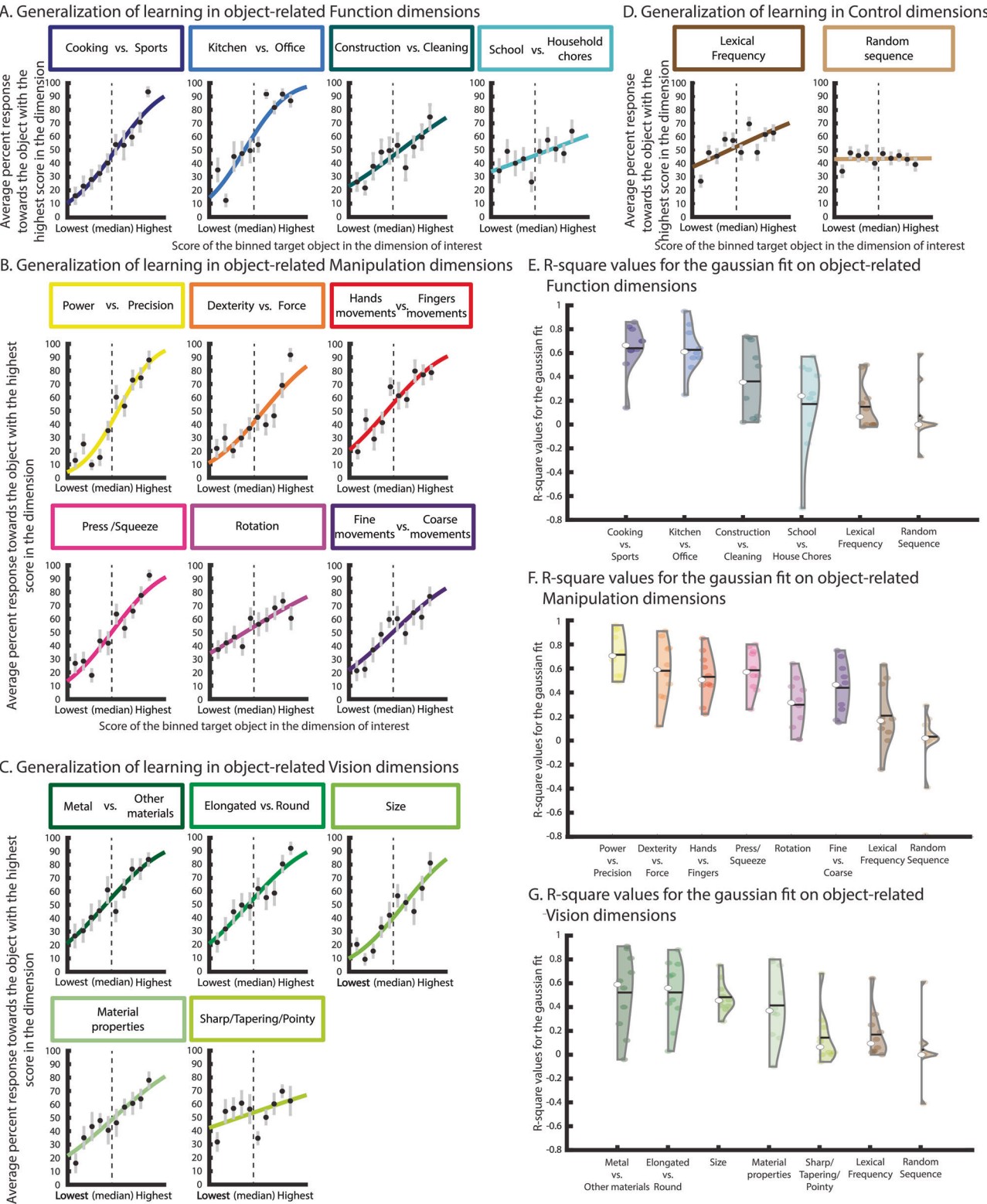

**Fig. 3 Supervised learning of object-related dimensions in a categorization task. A–D** Here we show percent response towards the object with the highest score for each of the object-related dimensions and the two control dimensions (i.e., towards the extreme with the highest score). Percent responses were averaged within each of the ten bins, and a cumulative Gaussian curve was fitted on the data of each individual. The presented plots are based on the average of all participants. Error bars correspond to SEM ($N = 10$ participants per dimension in a total of 270 participants); depicted cumulative Gaussian curve was fit on the average percent results for visualization purposes; **E–G** Violin plots of the R-square values of the Gaussian fit for each dimension for each participant.

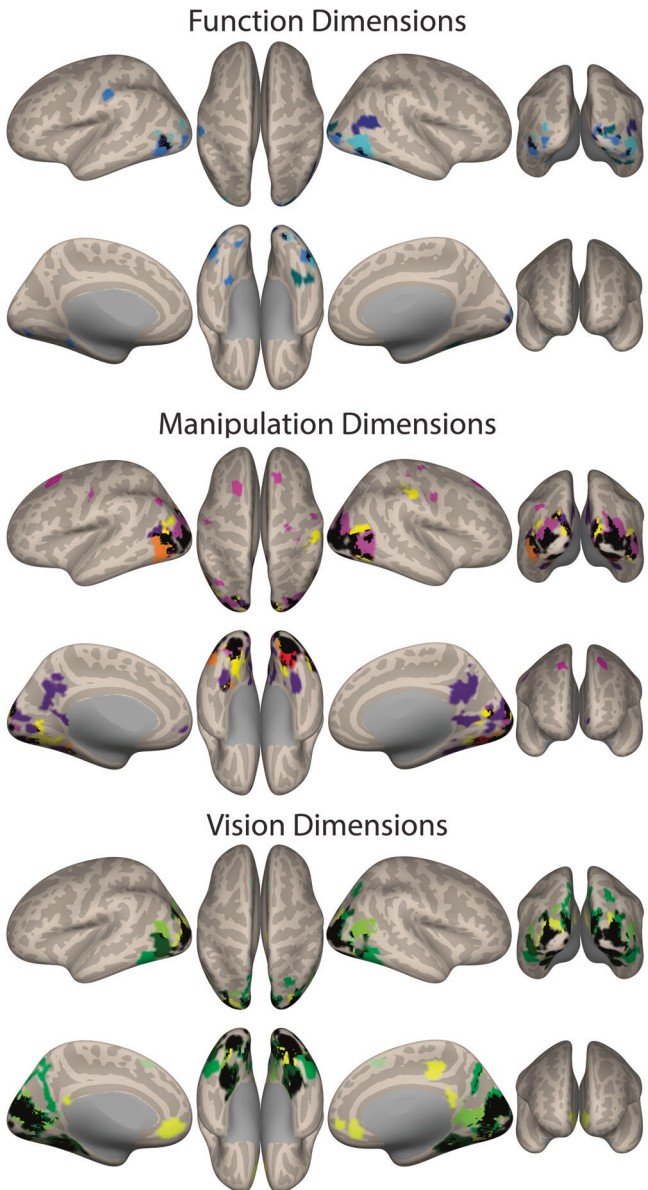

**Fig. 4 Neural effects of the object-related dimensions.** Here, we show an overlap map with all object-related dimension (F-maps per dimension against zero – i.e., against no modulation) per knowledge type, each dimension color coded by the colors in Fig. 3. Black corresponds to areas where at least two individual dimensions overlap (all individual F-maps cluster-forming height-thresholded at $p < 0.001$ – except for the dimensions "Cooking vs. Sports" and "Material properties" that are cluster-forming height-thresholded at $p < 0.005$ – and all corrected at FDR $p < 0.05$; all $N = 26$ participants).

around major action goals and the contexts in which objects are usually encountered. Moreover, these dimensions were able, collectively, to explain neural responses in pMTG/LOC, as these regions code for object-related action knowledge and meaning[53–56,58]. A particular result obtained in our parametric modulation analysis relates to a superior-to-inferior dichotomy within pMTG/LOC in the explanatory power of the first two function dimensions. The "Kitchen vs. Office" dimension – putatively related with context – explained activations in more inferior aspects, whereas the "Cooking vs. Sports" dimension – more related with particular types of activity – explained more superior activation. Interestingly, pMTG/LOC has been shown to

be topographically organized in a superior-to-inferior gradient that codes for sociability versus transitivity[58] – perhaps speculatively, cooking and playing sports may reflect more action-related sociability than the context in which actions are performed (kitchen or an office). Related with this, and again as predicted, activity within parahippocampal gyrus, putatively coding for spatial relations between objects and layout information[22,30], was modulated by object scores in the context-related "Kitchen vs. Office" function dimension. There were, however, a couple of effects that were not expected for the function dimensions. Specifically, the dimension "Cooking vs. Sports" significantly explained activation in right lingual gyrus extending superiorly to the cuneus. Interestingly, studies that focus on the mere presence of other objects during a transitive action (i.e., contextual objects), and the amount with which these objects relate to the goal of the actor, lead to activations around the regions obtained here[57,100,101], which could arguably be what leads to a modulatory effect in this region for this dimension. Finally, signal in the left IPL seems to be modulated by the scores of the "Kitchen vs. Office" function dimension. This region is known to be important for praxis and object manipulation[14,44,45,51,79,80], and may require information about function for accessing the full motor program associated with a target manipulable object[14,16,17]. Thus, the fact that object function is needed in order to access object-specific motor programs may potentially explain our results.

The content covered by manipulation dimensions also conforms, in part, to what was predicted, in that we obtained dimensions that relate strongly to object affordances – in particular, the type of grasp afforded by the manipulable objects used – and to motor aspects that are not necessarily directly available from the inspection of an object (e.g., "Rotation"). Thus, posterior parietal cortex and dorsal occipital cortex, including V3A, IPS (extending to post-central gyrus), and Precuneus – areas involved in hand shaping for grasp, grasp planning, processing different types of grip formations[32,44,59,60,62,64,90,102–106] (but see refs. [60,61]), and coding for shape properties and 3D representations of objects independently of action[70,72–75,82,89,107] – were predictably explained by manipulation dimensions, and especially the "Power vs. Precision" dimension. Notably, several aspects regarding the manipulation dimensions were not in line with our initial expectations. Firstly, one of the major dimensions of manipulation – "Dexterity vs. Force" – may be conceived more of as a dimension that is important for interacting with an object that is less motor-related than the other dimensions obtained. In fact, and perhaps speculatively, the involvement of ventral regions under this dimension (both posteriorly within lingual gyrus, and anteriorly within parahippocampal gyrus), is consistent with the importance of surface and material properties in deriving the weight of an object[108], and thus the amount of force or dexterity needed to act with these objects[109]. Secondly, we predicted manipulation dimensions would explain responses in more anterior areas of parietal cortex, and namely the left IPL, as this region is causally involved in the retrieval of praxis[14,44,45,51,79,80]. Nevertheless, none of the manipulation dimensions obtained was able to explain responses within left IPL. This unexpected result may reflect the fact that the dimensions obtained relate to piecemeal aspects of the full-blown motor program associated with an object (e.g., grasp, wrist rotation), whereas left IPL seems to be engaged when retrieving an object's unique full motor program[51,80].

Finally, regarding visual dimensions, we did observe the predicted major dissociation between those dimensions that relate to geometric properties (e.g., dimensions "Elongated vs. Round" and "Size") and those that mirror material properties of objects (e.g., dimension "Metal vs other materials"). This dissociation was also

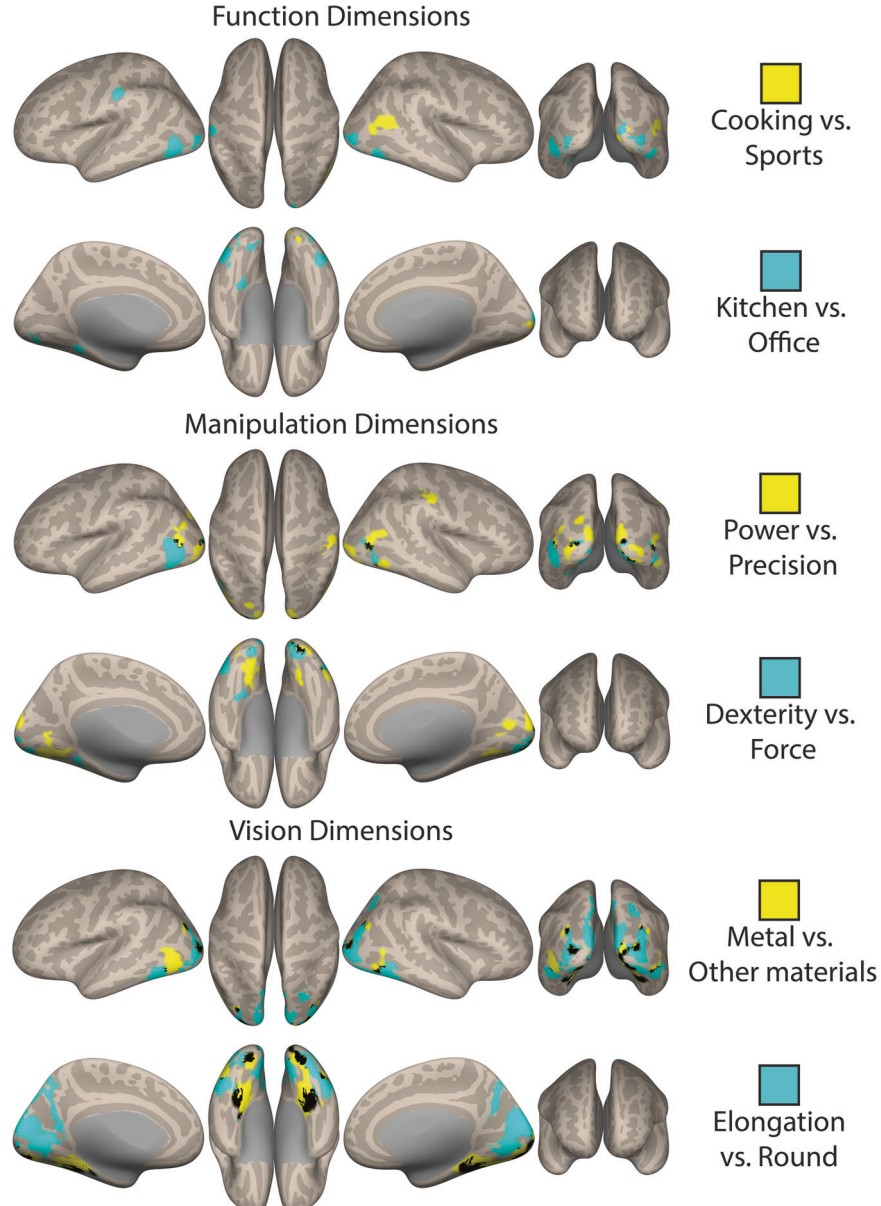

**Fig. 5 Content specificity in the neural responses.** Here, we show modulatory effects (i.e., the beta values of each dimension that are significantly different from zero) of the first two dimensions of each knowledge type (all individual F-maps cluster-forming height-thresholded at $p < 0.001$ – except for the dimension "Cooking vs. Sports that is height-thresholded at $p < 0.005$ – and corrected at FDR $p < 0.05$; all $N = 26$). In yellow we present the first dimensions of each knowledge type ("Cooking vs. Sports"; "Power vs. Precision"; "Metal vs. Other Materials"), whereas in light blue we present the second dimensions of each knowledge type ("Kitchen vs. Office"; "Dexterity vs. Force"; "Elongation vs. Round"). Each map shows voxels where signal is explained by the dimension when compared to no modulation (i.e., when compared to 0). Black corresponds to areas of overlap between the two dimensions presented per map.

reflected in the fact that the neural responses explained by the scores within each dimension followed the lateral-to-medial occipito-temporal gradients for geometric and surface related content[82–84]. Specifically, the largest clusters of activation for the dimension "Metal vs. other materials" are medially along the collateral sulcus (posterior to anterior) bilaterally, para-hippocampal gyrus and the posterior lingual gyrus. As discussed, these regions have been associated with the processing of material and surface properties of objects[82–85]. The dimension "Elongated vs. Round" shows a major cluster of explained signal in more lateral of ventral temporal cortex. Specifically, LOC is a center for shape processing, and its disruption severely affects shape processing[82–84,86–88]. Moreover, this dimension strongly

explained neural responses within aspects of dorsal occipital cortex. As a group, both these lateral and ventral temporal, as well as dorsal occipital regions are exactly those that have been shown to be coding for object roundedness and elongation[19,69,89–91].

One important aspect to discuss relates to whether these dimensions (especially dimensions from different knowledge-types) correlate in our environment and in the results reported herein. Indeed, potential correlations between such dimensions are not only a collateral effect of the complexity of the world that surrounds us, and of information that is not strictly sensory, but are also what potentially defines the nature of object-related information – i.e., that it is multimodal, complex, mid-to-high level object information. Importantly, a small number of our

dimensions do show moderate-to-high correlations with one another (see Fig. S5). This may be a potential limitation of this study, in that these correlations may mean that there is shared information between the dimensions. However, and perhaps most importantly, these correlations are a reflection of real-world natural co-occurrences – e.g., most kitchen utensils are made of metal, and the correlation between these two dimensions reflects that. Moreover, there is also a large degree of independence between even the most correlated dimensions, suggesting that this sharedness of information between the dimensions may be minimal.

Beyond the neural and content-specific results described above, a major aspect of our data relates to the ability of these dimensions to predict behavioral categorization judgements. Specifically, we trained participants to categorize objects while implicitly using a metric based on (one of) these dimensions. Critically, we showed that their learning could be generalized to untrained items (cf. the lack of generalization for our control dimensions). These data strongly suggest the validity of these dimensions in organizing the conceptual space of manipulable objects, as well as that of our approach to understanding the multidimensionality of object space. However, the use of a learning paradigm, rather than a less constrained paradigm where participants were free to use any dimension, is a limitation of our study, and may raise doubts as to the centrality and spontaneity of these dimensions to manipulable object processing. We believe, however, that in order to understand the role of each of these dimensions for object categorization and processing we needed to direct participants, implicitly, to our target dimensions, given the multitude of dimensions we obtained per knowledge type. Moreover, some of our data may help address this limitation. Specifically, we do not obtain generalization effects for the lexical frequency control dimension – a major linguistic dimension. Furthermore, we see a clear trend for stronger generalization effects for the dimensions that show higher stress values in the MDS solutions. These facts, and particularly the latter one, strongly suggests that specificity in the learning and generalization to novel objects is unique to the key object-related dimensions and not just to any dimension, and thus that these dimensions differentially structure how we think about manipulable objects. That is, transfer of the learning is not uniform but rather correlates with how central that dimension is for understanding how we represent objects in each knowledge type.

Finally, not only is our data in line with what has been obtained when trying to understand overall multidimensionality of object processing[13,40,110], but they also further our understanding of the fine-grained properties of object processing that was lacking hitherto. Several approaches have demonstrated, as we have here, that multidimensionality is central for object processing[13,40,42]. These studies have suggested a series of dimensions that underlie object representations. Interestingly, when we look at some of the individual dimensions proposed by these groups (e.g., round; sport-related[13]) they do relate to those that we have obtained here. Moreover, the nature of the dimensions reported here potentially relates to mid-level object properties – and thus are probably similar to what Fernandino and collaborators[42] call "experiential features" (notwithstanding potential differences in the format of these representations; e.g., modal or amodal). However, one aspect that differentiates these previous research efforts and what we present here is that we focus on within-category processing. Interestingly, some (perhaps even most) of the dimensions obtained by these authors do relate to domain (e.g., Tool-related; Animal-related; Body-part-related[13]). This brings about two central issues: 1) that domain seems to be a major principle of organization; and 2) that the presence of these categorical dimensions may exhaust much of the variance present

in the data[13,40,42]. Overall, these issues call for within-domain scrutiny of the multidimensionality of mental representations and object processing. Here, we went further and explored the finer-grained details of object processing at a within-category level, and showed content-specific dimensions that guide the way we perceive manipulable objects beyond (and in a way independently of) the macroscopic differences between domains. These content-specific dimensions are central to the representations we build of the objects we perceive. We can then use these representations in the process of identifying objects and compare them to the object representations we have stored in our long-term memory.

Overall, then, we show that object-related dimensions, extracted from the subjective understanding of a large group of individuals over a large set of manipulable objects, can guide our behavior towards these kinds of objects and can explain manipulable object-specific neural responses, and potentially the finer-grained organization of object-content in the brain, suggesting that multidimensionality is a hallmark of neural and conceptual organization.

## Methods

**Participants**. A total of 339 individuals (305 women) from the community of the University of Coimbra participated in the experiments (age range: 18–41): 60 in the object similarity sorting task, 43 in the label generation task (23 were presented with words and 20 with pictures), 270 in the supervised learning categorization task, and 26 in the fMRI task. All experiments were approved by the ethics committee of the Faculty of Psychology and Educational Sciences of the University of Coimbra, and followed all ethical guidelines. Moreover, participants provided written informed consent, and were compensated for their time by receiving either course credit on a major psychology course or financial compensation. All ethical regulations relevant to human research participants were followed.

**Stimuli**. We first selected a set of 80 common manipulable objects (see Table S1 for all the objects; see Figure S1 for examples of the images used). These were selected to be representative of the different types of manipulable objects that we use routinely. For Experiments 1 and 3 we used words to represent the object concepts, whereas for Experiment 4 we used images. We used both images and words separately for Experiment 2. Images were selected from the world wide web. We used an imaging editing software to crop the images, extract any background, and gray scale and resize the images to a 400-by-400 pixels square. Presented images subtended approximately 10° of the visual angle. We selected 10 exemplars per object type in a total of 800 images. For Experiment 4, we additionally selected 20 images of animals to function as catch trials.

**Similarity ratings and dimension extraction**. We first obtained similarity spaces for the different object-knowledge types tested (visual information, functional information and manipulation information). We presented participants with words referring to our 80 manipulable objects and asked them to think about how similar these objects were in each of the knowledge types (e.g., function). We used an object sorting task to derive dissimilarities between our set of objects[111] Fig. 1A), because sorting tasks have been shown to be a highly efficient and feasible way to obtain object similarities from large sets of objects. For each knowledge type, each participant was asked to sort all 80 objects into different piles such that objects in a pile were similar to each other, but different from objects in other piles, on the target knowledge type. Each participant went through the three knowledge types independently (order counterbalanced across participants) – that

is, each participant was asked to judge the similarity of the objects 3 different times in a row (one for each knowledge type). We used words instead of pictures to avoid sorting based on exemplar-specific similarities. The sorting was performed in Microsoft power point. The final sorting was saved and later analyzed.

**Label generation for extracted dimensions**. To test for interpretability of the extracted dimensions, we asked participants to generate labels for the 15 key dimensions. Each participant went through all of the 15 key dimensions (order of dimensions randomized for each participant). For each dimension, participants were presented with the 20 most extreme objects of that dimension – 10 pertaining to each one of the extremes of the target dimension – rank ordered by the value on the target dimension. Pictures or words were presented such that in one side of the screen we lined up the 10 objects of one of the extremes, and on the other side of the screen the other 10 extreme objects of the dimension. At the center of the screen, we presented ellipsis between parenthesis to convey continuation and the presence of other objects in between. Participants were then told that they could generate up to 5 labels per dimension that, in their view, best explained the difference between the objects at the two extremes (see Fig. 1B). Participants either saw images of the objects or words referring to the objects throughout the experiment. Participants were told to which object knowledge the dimensions belonged to (i.e., vision, manipulation, or function). We used these two modalities of presentation to avoid presentation-specific results. We focused on which labels were more consensual across participants in the label generation task by collating the labels generated by the participants, and analyzing frequency of production of each label (see Fig. S3 for the labels generated for each dimension).

**Supervised learning of object-related dimensions**. In this task, we taught participants to categorize a subset of our 80 objects in terms of their scores along a target dimension, and then tested whether their learning could be generalized to a subset of untrained (i.e., the remaining) items. We first divided the dimensions into 10 bins of 8 objects each, defined according to the values of the target dimension. We used the most extreme objects at each end of the dimension (i.e., those with the highest and the lowest score in the dimension) as anchors in the categorization tasks. Per trial, participants were asked to categorize, as fast as possible, the presented target objects (in a word format) as to whether they were closer to one or the other extreme object of the target dimension (e.g., pepper grinder or horn for the function dimension "Cooking vs. Sports"). To do so, participants had to press the right or left buttons of a button box – response assignment was randomized across participants. Participants were not told the labels of the target dimension, and were only informed as to whether the dimension was related to vision, function or manner of manipulation of the target objects.

The experiment was divided in three phases (in Fig. 1C, we show the different phases of this experiment). In the first phase, we wanted participants to learn to associate objects with high or low dimension scores with the correct extreme. Thus, we selected 4 items from each of the 4 most extreme bins (i.e., bins 1, 2, 9, and 10), and 2 from each of the third most extreme bins (i.e., bins 3 and 8), in a total of 20 objects – i.e., we selected objects that were most strongly related with the two extremes of the dimension in order to establish a robust understanding of the target dimension. These were selected randomly within each bin per each participant. In each trial of the first phase of the experiment, participants were first presented with the words referring to the two extreme objects in the upper corners, according to the

response assignment, and a fixation cross for 500 ms. We then presented the word referring to the target object at fixation, along with the words referring to the two extremes of the target dimension (in the upper corners of the screen). This remained on the screen for 2.5 s after which the correct response was presented right below the target object. Specifically, participants were presented with a sentence that said that the object presented was closer to one of the extremes. This sentence remained on screen for another 2.5 s. Participants were asked to pay attention to these sentences and learn the assignments between objects and extreme anchors. Each of the 20 objects was repeated three times in a total of 60 trials. Thus, in phase one, participants were not required to respond, but just to learn the associations of each of the presented objects with the extremes of the dimension.

In phase two, we wanted participants to continue learning the associations between target objects and the extreme anchors, and also to extended the learning set to all bins. Thus, we selected 5 items from each bin – including the 20 items used in phase one. In this phase, participants were presented again with the target object (3 repetitions, in a total of 150 trials), but this time were required to respond and categorize the target objects as being close to one of the extremes (e.g., closer to the pepper grinder or the horn). After responding, participants were given feedback as to whether they were correct or incorrect. The trial structure was in all similar to the one in phase 2 except that the object was presented for 2.5 s or until a response was obtained, and this was immediately followed by feedback as to whether the response given was correct. The feedback stayed on screen for 2 s.

Finally, in phase three, we wanted to test whether the learning could be generalized to untrained items. Thus, all items were used in this phase of the experiment (i.e., the 50 trained items and 30 untrained items – 3 from each of the ten bins) and were repeated 6 times (in a total of 480 trials). This phase of the experiment was in all equal to phase 2, except that there was not feedback given – that is, after categorizing a target object as to whether it was closer to one or the other extreme of the target dimension, participants would start the next trial.

All 15 key object-related dimensions were tested in this experiment (each participant as tested on only one dimension; 10 participants took part in the experiment for each dimension). Moreover, we added two control dimensions. In one of these controls, we took one of the real dimensions (the first function dimension) and randomly shuffled the scores of the dimension for the objects. For the other control, we took lexical frequency values[112] for each of the objects and rank ordered them in terms of these values. Each of these controls was run in 30 participants (i.e., 10 associated with each knowledge type). We used these dimensions to control for reliable generalization of object-related dimensional learning to untrained items.

**fMRI object categorization task**. The fMRI task consisted of 1-3 sessions spread across separate days (due to the onset of COVID-19 pandemic, only 19 subjects completed all 3 sessions, with 5 & 2 remaining subjects completing 2 & 1 sessions, respectively). Each session contained 3 event-related runs (duration: 456 TRs (repetition time; 912 s), resulting in 3-9 completed runs per subject. In each run, subjects centrally fixated gray-scaled images of: 1) Manipulable objects (160 trials per run; 1 exemplar for each of the 80 objects was randomly selected from a larger stimulus set of 800 images (80 object identities x 10 exemplars each), and presented twice per run; and 2) 'Catch' animal stimuli (8 trials per run; 8 unique animal exemplars randomly drawn from a set of 20 images). Trial length was 4 s (2 s image presentation + 2 s fixation) and 54 null events (4 s fixation) were also included in the design. Because each object image was presented twice per run,

randomization of trial order was performed for the first and second half of each run separately (i.e., to avoid strong recency effects where some items may, by chance, be repeated within a much smaller time period than others), where 80 object stimuli and 50% of catch trials and null events were presented in each half. Subjects were instructed to maintain fixation continuously (a fixation dot appeared in the center of the screen for the entirety of the run) and to make a simple button-press judgment for each image trial (object or animal).

**MRI acquisition**. Scanning was performed with a Siemens MAGNETOM Prisma-fit 3 T MRI Scanner (Siemens Healthineers) with a 64-channel head coil at the University of Coimbra (Portugal; BIN - National Brain Imaging Network). Functional images were acquired with the following parameters: T2* weighted (single-shot/GRAPPA) echo-planar imaging pulse sequence, repetition time (TR) = 2000 ms, echo time (TE) = 30 ms, flip angle = 75°, 37 interleaved axial slices, acquisition matrix = 70 × 70 with field of view of 210 mm, and voxel size of 3 mm$^3$. Structural T1-weighted images were obtained using a magnetization prepared rapid gradient echo (MPRAGE) sequence with the following parameters: TR = 2530 ms, TE = 3.5 ms, total acquisition time = 136 s, FA = 7°, acquisition matrix = 256 × 256, with field of view of 256 mm, and voxel size of 1 mm$^3$.

**Statistics and reproducibility**. The size of each sample corresponds to typical samples sizes of these kinds of experiments (e.g., see ref. [33]).

*Dimension extraction*. In Fig. 1A, we show how data extracted from the piling task was used to produce dissimilarity matrices. Per participant, we obtained three (vision, function, and manipulation) 80 by 80 dichotomous (0 and 1) matrices that coded for membership of each pair of objects to the same sorting pile (i.e., if objects $i$ and $j$ were on the same pile, the value of the cell $ij$ was 1, else it was 0). These individual matrices were then averaged over the participants and transformed into 3 final dissimilarity matrices – one per knowledge type.

The dissimilarity matrices were analyzed with the use of non-metric multidimensional scaling using Matlab. The number of dimensions to be extracted was determined based on *stress* value (Kruskal's normalized stress[93] – i.e., the fit between the distances among the objects in the dimensional structure obtained with n-dimensions, and the scores in the input matrices. We obtained stress values for the MDS solutions with different numbers of dimensions. Stress values below 0.1 are considered acceptable[93,113], suggesting that the dimensional solution, and estimated distances between objects, reasonably fits with the dissimilarities from the input matrix, while still imposing good dimensionality reduction. As such, we selected the first solution below stress values of 0.1 (see Fig. S2 for scree plots for each of the knowledge types).

*Analysis of supervised learning data*. To analyze the data of phase three, we first averaged responses per bin (3 untrained items repeated 6 times). For each participant and dimension, we plotted the percent responses towards the object with the highest score along that dimension as a function of the ten bins and fit a cumulative Gaussian to these data following the equation:

$$y = \frac{1}{2} erfc\left(\frac{-\sigma(x-\mu)}{\sqrt{2}}\right)$$

where y is the percent responses, x is percentile bins, σ is the slope of the cumulative Gaussian, μ is the midpoint of the curve, and

erfc is the complementary error function. The cumulative Gaussian spans reports from 0% (lower asymptote) to 100% (upper asymptote). We expected that if the object-dimensions were cognitively important for our ability to process and recognize objects, then participants would be able to generalize their learning of the target dimensions to untrained items – thus, percent responses towards the extreme object with the highest score should increase as a function of the increase in the dimensional score per bin. R-square values per participant were obtained as a measure of goodness of fit between the cumulative Gaussian curve and each participant's data. We used a liberal R-square cutoff value of 0.30. This specific cutoff is necessarily arbitrary, but we used this liberal criterion to allow us to determine if any given dimension is reasonably predictive of each participants' behavior.

*MRI preprocessing, and analysis*. Data were preprocessed with SPM12 (i.e., slice-time correction, realignment (and reslicing), anatomical co-registration and segmentation, normalization, and smoothing). Analysis was performed on smoothed, normalized data (normalized to MNI template; 3 mm isotropic voxels). General linear model (GLM) estimation was performed in SPM12 (data were high-pass filtered (256 s) and a first-order auto-regression model (AR(1)) was used to estimate serial time-course correlations). For each subject, a GLM was estimated separately for each knowledge type because mathematically these dimensions are necessarily uncorrelated with each other (i.e., MDS produces uncorrelated dimensions). Each dimension/modulator was first scaled to an interval of 0-1 (due to the large differences in original scales of the key dimensions (ranging between −0.42 to 0.65) and then mean-centered, as is typical for parametric analysis[114]. We ensured that serial orthogonalization of modulators was not implemented, so that all modulators would compete equally for model variance (rather than in the case of serial orthogonalization, assigning all shared variance to the first modulator, with subsequent modulators competing for the remaining unexplained variance, and therefore strongly biasing effects towards the first modulator, at the expense of all others; note that the lexical frequency dimension was also included as a modulator and as such, key dimensions only account for variance not explained by lexical frequency). The following regressors comprised the design matrix for a given run: 1 regressor for manipulable object stimuli (i.e. box-car regressor for all tool stimulus events, convolved with the SPM canonical haemodynamic response function), N key dimension modulators (e.g. vision dimensions 1-5), 1 lexical frequency modulator, 1 catch stimuli regressor (all catch stimuli; modeled but not analyzed), N 'placeholder' modulators for the catch regressor (i.e. a matched number of dimension modulators were required here for design balance, but these had no modulatory effect as all values were set to zero), 6 head-motion regressors (plus an intercept regressor at the end of the full design).

The resulting beta maps for each of the dimensions describe the extent to which object stimulus responses vary as a function of item position along a particular dimension. Importantly, because the obtained dimensions reflect an arbitrary directional ordering of items, and therefore are not inherently uni-directional (i.e., across repeated non-metric MDS solutions (with random initializations), relative item-to-item distances are stable but the overall item ordering can be reversed in either direction), both positive and negative slope effects here reflect sensitivity to a given dimension, based on either possible directional ordering. Thus, contrast images per dimension (contrast vector with 1 for the dimension modulator, 0 for all other regressors) were then entered into a group-level F-test to accommodate 2-tailed effects (i.e., positive or negative slopes that differed from zero). F-maps

were FDR cluster-corrected ($p < 0.05$) with a height-threshold of $p < 0.001$ (but due to limited coverage for the first function dimension and fourth vision dimension, a slightly more lenient height-threshold of $p < 0.005$ (FDR cluster-corrected $p < 0.05$) was used, as previously motivated (e.g., see refs. [115–117]). In short, the resulting F-maps show which brain areas demonstrate modulation sensitivity to each specific dimension.

For easy visualization of brain coverage associated with the dimensions of a particular knowledge type (see Fig. 4), a composite dimension map was generated where supra-threshold voxels for each dimension were coded with a unique integer/color (and voxels with overlapping coverage from 2+ dimensions were coded with a different integer/colored black) Similar composite maps for the first 2 dimensions of each knowledge type were also created (see Fig. 5). All maps were corrected for multiple comparisons (cluster-forming threshold $p < 0.001$ – or $p < 0.005$ for "Cooking vs. Sports" and "Material properties" – and FDR correction threshold $p < 0.05$). All maps were projected to an inflated surface with the CONN toolbox[118].

For reproducibility we provided the behavioral and fMRI data obtained. See data availability statement below.

**Reporting summary.** Further information on research design is available in the Nature Portfolio Reporting Summary linked to this article.

## Data availability

All data can be found at https://osf.io/jzuf3/ (https://doi.org/10.17605/OSF.IO/JZUF3)[119].

## Code availability

Custom code used can be found at https://osf.io/jzuf3/ (https://doi.org/10.17605/OSF.IO/JZUF3)[119].

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

## Acknowledgements

This work was supported by the European Research Council (ERC) under the European Union's Horizon 2020 research and innovation programme Starting Grant number 802553; "ContentMAP" to J.A; A.F. is supported by a grant from the Biotechnology and Biology research council (BBSRC, grant number: BB/S006605/1) and the Bial Foundation, Bial Foundation Grants Programme Grant ID: A-29315, number: 203/2020, grant edition: G-15516. S.K. is supported by a Fundação para a Ciência e Tecnologia (FCT) Doctoral Grant SFRH/BD/145218/2019. DV is supported by a FCT Doctoral Grant SFRH/BD/137737/2018. F.B. is supported by a FCT Individual grant CEECIND/03661/2017. J.W. is supported by a FCT Individual grant CEECIND/03185/2021. The authors wish to thank Bradford Mahon for his comments on an earlier draft.

## Author contributions

J.A. J.W., D.V., and S.K. prepared and conducted the experiments; J.W., J.A., A.F., F.B., and R.C. analyzed data; J.A. prepared the manuscript; J.A. A.F., S.K, D.V., F.B., R.C., Z.T., and J.W. reviewed the manuscript. J.A. obtained the funding, and conceived the project; J.A. and J.W. supervised the project.

## Competing interests

The authors declare no competing interests.
