## [Peer Review File · Communications Biology]

Reviewers' comments:

Reviewer #1 (Remarks to the Author):

(1) The authors present the results of their study as if they showed that people spontaneously and naturally use the dimensional fine structure found within certain object categories (visual shape, manipulation type, and function). I think that this interpretation is a bit over the target, as I will explain in the following. In the first step, the authors pre-specified three categories in the Multidimensional Scaling Test, for each of which 80 objects were separately assessed for similarity. A second group of subjects then generated labels for the identified dimensions within each of the three categories. A third group of subjects was implicitly trained in a supervised-learning categorization task to categorize objects along the now established dimensions. Also, feedback was provided in this multilevel training. Here, the authors emphasize that in the end even untrained objects could be successfully categorized along the established dimensions. Contrary to the authors' claim, however, this result does not show that the identified dimensions are of central importance for the organization of object knowledge. It only shows that people acquire implicit knowledge about object (or more generally: item) categories and can apply it to untrained objects (items). In principle, all conceivable object properties can be described and trained dimensionally, without telling us anything about the naturally existing mental or neural representation of objects. Even if the categorization behavior of the third group of subjects is based on implicitly acquired knowledge, it is still based on acquired knowledge and not on spontaneous behavior. In short, learnability says nothing about relevance or importance. The fact that different groups of participants were used does not change this conceptual restriction. The problem of reasoning is exacerbated by the fact that the entire construct is based on pre-specified categories. This decision is based on (certainly reasonable and plausible) presuppositions about how object knowledge is represented; but they remain presuppositions, so that all further conclusions must always be considered with this limitation. The rationale for the authors of this study choosing a training design is found somewhat late and rather incidentally in the statement "We used a training task [...] as there were several dimensions per knowledge type that could be used by the participants if we opted for an untrained experimental situation". This reasoning is absolutely plausible, but unfortunately it also spoils the conclusion that can be drawn from the data obtained. These limitations should be clarified in the manuscript, and authors should avoid over-interpretation.

(2) Using these dimensions to model brain activity in a different set of subjects is a compelling approach. Since the task of these subjects was an object-animal categorization, the results regarding the effects of the dimensions appear to be valid. A criticism of the fMRI approach and analysis is that it would have been more plausible to use an RSA rather than a univariate approach via parametric modulators reflecting the dimensions used. Authors should explain why they did not choose the former approach.

(3) At first glance, the labels of the dimensions seem to help make neural representations interpretable. In fact, however, exactly this transfer is not - and I would also say: cannot be - provided by this study. For example, what exactly does it tell us that cooking objects activate the right pMTG more than sports objects do? One could speculate in what ways cooking and sports objects differ, and it is evident that they do so on many different (even more fine-grained) dimensions. But which of these many options explains the neuronal differences remains unaddressed. I would not say that it is not an interesting or valid finding that the right pMTG is more active for cooking objects, but it does not lead us to a better understanding of what exactly rules the (presumably multidimensional) neural representation of objects. The authors repeatedly emphasize that, in contrast to previous studies, they choose a finer within-category approach instead of distinguishing between (coarse) categories. In doing so, however, they seem to forget that the finer within-category dimensions were dependent on the coarser pre-selected categories. Thus, the question of object representation is possibly only shifted to another level, but not answered fundamentally differently: We see the same pattern of regional differences along dimensions as well as we can identify these differences at the

categorical level. However, the nature of these differences - see the example of cooking vs. sports objects - is not clarified. Interestingly, in the final statement of their discussion, the authors arrive at a more realistic interpretation of their results. It would be important that speculative and over-interpretative parts of the discussion are reduced appropriately.

(4) The discussion of the fMRI findings seems largely post-hoc and addresses a variety of brain regions. This gives the results something arbitrary. Even if several of the regions discussed later are mentioned in the Introduction, it is nevertheless clear that an exact assignment in the sense of a hypothesis testing did not take place and was probably not the most important goal of the study. Since the dimensions were derived from behavioral results in a first step, one could (but did not) have formulated neurofunctional hypotheses for these dimensions. I suggest that the post-hoc character should be explicitly pointed out. Also, parts of the discussion are starting already in the results section, something seems to have gotten mixed up there. Finally, the discussion is really too lengthy and often there are redundant paragraphs, so it can and should be shortened.

Reviewer #2 (Remarks to the Author):

This paper is designed to uncover the nature of object knowledge, with specific focus on within category organization as many studies have mapped between category dimensions already. First, the authors acquired human subjective judgments on a large set of manipulable objects on several dimensions i.e., vision, function and manipulation which were explainable and generalizable. They then uncovered neural regions associated with the dimensions and conclude that the same dimensions govern the neural profile primarily in areas with manipulable-object preferences compared to other categories of objects.

The approach adopted is interesting but there are several concerns which undermine this reviewer's enthusiasm, some of which are methodological, some more theoretical.

a. It is interesting that there was a major divide in the neural correlates of the two dimensions with material properties represented in more medial ventral temporal cortex, and in FG and LO. Are there a priori reasons why this should be? Suggesting that this might be related to mid-level properties is not a strong enough explanation unless you can demonstrate this is in some way otherwise this seems more a re-description of the findings.

b. The analysis of the imaging data requires strengthening. The data are all analysed in terms of which regions are significant for which dimension (related to question in a). Seems that a better approach is to derive measures of the relationship between the dimensions and the neural signal. The analysis is by fiat - there is activation so this must be significant. Seems to be the wrong way around? More detailed examination and exploration is required.

c. The behavioral data are, in general, stronger than the neural data but I do not understand why there is training? If the knowledge base of observer is already configured along these dimensions (which it should be if this is the basis of object knowledge). The authors state that "as there were several dimensions per knowledge type that could be used by the participants if we opted for an untrained experimental situation". But isn't the whole point to establish the dimensions? Much of the paper is presented as though this is the goal. Otherwise, the findings seem to show that when one trains participants, they learn? Okay, the dimensions do differ but this does not indicate the native knowledge structure of the participant.

d. Do you think there is activation in other parts of cortex aside from the areas included in Figure 4 composite dimension map. In its current form, it is misleading and would be better to have reader see the entire distribution of activation for each dimension. Is it indeed the case that when you plot F map against zero, there is no other activation? How would you reconcile this with the very many results that show dorsal cortex activation for affordance from manipulable objects? Or anterior temporal lobe for functional knowledge?

Reviewer #3 (Remarks to the Author):

Disclaimer: I was one of the reviewers at a previous journal. I appreciate the authors addressing many of the issues mentioned in the previous round of reviews, specifically regarding the framing of the work. I still see several outstanding issues that I think would be important to address.

The main motivation of the authors has become a lot clearer through the revision, and the framing with a focus on manipulability (i.e. within-category) is much more in line with the results. I would appreciate if the authors were a bit more consistent with this framing, since many of the original formulations continue to be present in the manuscript that do not directly appear to follow from the results, specifically in the abstract but also in the main text (e.g. "multidimensionality is a signature of high-level object processing" or "perhaps in the same way as the different dimensions that rule the organization of low-level sensory-motor cortices overlap spatially, so do dimensions that rule the organization of object knowledge in the brain."). I understand that they aim at drawing more general conclusions but this alternative framing either appears too early in the manuscript without justification, or could be better justified later.

Currently, it is also not entirely clear whether the focus of the work is on multidimensionality as such, which is then confirmed by the results in the brain, whether the focus is on identifying multidimensionality in the brain, for which the behavioral results are just a necessary element, or whether it is both (behavioral and neural representations). It would be good if the authors could frame this more consistently.

For the interpretability task, one issue that I currently do not see addressed is whether participants knew which knowledge type they were assessed on (visual, function, manipulation). It just appears to be striking that participants could tell the knowledge types from each other so clearly, and it would be worth adding an analysis of this to the interpretability results if this truly was the case. It is, of course, fine if they were instructed of the knowledge type, even though this may somewhat limit the interpretability.

More generally, it would be very instructive to see how strongly the different dimensions are related with each other between knowledge types to better understand their relationship. If they are strongly correlated, they may be drawing on similar information. If they are weakly correlated, this would indicate a clear separation of information between knowledge types.

One additional reason why this may be relevant is the fact that the current brain-based analyses are executed separately for different domains. If these behavioral dimensions are strongly correlated between domains, then it might be good if the authors discussed this as a potential limitation of the interpretability of their results.

In the learning task, is it possible participants solved the task merely by asking which object is more similar to the reference object with respect to function, manipulation, or vision? Since participants had been instructed, this alternative approach may, in fact, be quite simple yet would not indicate interpretability of dimensions but merely the ability to report similarity to two reference objects. Perhaps the authors could run an analysis highlighting what the best possible performance would look like under this scenario.

Evaluating the overlap between clusters in a cluster analysis requires a conjunction analysis, specifically since a cluster surviving a cluster threshold can only be interpreted as "at least one voxel within the cluster shows information" (see Woo et al., 2014, Neuroimage, and Rosenblatt et al., 2018,

Neuroimage). Without this, the inferences drawn about spatial overlap of activations are statistically invalid.

The brain results could be made clearer, and there still seems to be a lot of reverse inference at play. For example, "many of these dimensions overlap spatially in particular regions" is a very vague statement. "most probably due to the fact that these dorsal occipital regions are involved in object-specific 3D processing" appears like a strong reverse inference. The entire section "Object related-dimensions maintain content specificity in their neural responses" appears to be trying to fit the results within a framework of existing findings, which leads to a lot of reverse inference. Given the diverse findings associated with object processing regions, it may be better if the authors focused more strongly on their key message: that multidimensionality explains findings and that it may also appear to explain other aspects of what we know about the functional organization in these regions. Thus, rather than relating the dimensions to previous findings, why not relate previous findings to the dimensions, thus highlighting their explanatory value, and thus turning a reverse inference into more of a forward inference? What are key findings in relation to manipulability of objects and how do these results capture these findings (or do not capture them)

Other comments:

- Given that the cutoffs chosen by the authors in Figure 3 continue to be arbitrary, are not justified by the authors, and do not seem to serve a clear purpose, why not remove them?
- There are still numerous typographic errors throughout the manuscripts (e.g. fine-grain instead of fine-grained). I think it would help if the authors had a thorough read through the manuscript to eliminate these.

Point-by-point responses to the Reviewers are presented below. We present the Reviewer's comments in **bold**, and our responses in normal font. All quotes from the revised manuscript will be presented underlined.

Reviewer #1:

(1) The authors present the results of their study as if they showed that people spontaneously and naturally use the dimensional fine structure found within certain object categories (visual shape, manipulation type, and function). I think that this interpretation is a bit over the target, as I will explain in the following. In the first step, the authors pre-specified three categories in the Multidimensional Scaling Test, for each of which 80 objects were separately assessed for similarity. A second group of subjects then generated labels for the identified dimensions within each of the three categories. A third group of subjects was implicitly trained in a supervised-learning categorization task to categorize objects along the now established dimensions. Also, feedback was provided in this multilevel training. Here, the authors emphasize that in the end even untrained objects could be successfully categorized along the established dimensions.

R: We want to start by thanking the Reviewer for all her/his work – it forced us to think deeply about the manuscript, the data, and the goal of the research. The comments and concerns raised by the Reviewer were central in our revision of the original manuscript, and we think the revised manuscript has benefited immensely from the thorough work of this (and the other) Reviewer(s). We hope that the Reviewer agrees that this version merits publication at Communications Biology.

1.1: Contrary to the authors' claim, however, this result does not show that the identified dimensions are of central importance for the organization of object knowledge. It only shows that people acquire implicit knowledge about object (or more generally: item) categories and can apply it to untrained objects (items). In principle, all conceivable object properties can be described and trained dimensionally, without telling us anything about the naturally existing mental or neural representation of objects. Even if the categorization behavior of the third group of subjects is based on implicitly acquired knowledge, it is still based on acquired knowledge and not on spontaneous behavior. In short, learnability says nothing about relevance or importance.

R: We thank the Reviewer for raising this concern. We completely agree that this issue had to be discussed in the manuscript. We have revised the manuscript to include a new paragraph in the discussion on the use of a training task, the limitations that the paradigm might bring, as well as our view as to why that limitation, although real, may be circumvented by the data obtained. Furthermore, we also tried to tone-down our discussion of the behavioral results given this limitation.

Specifically, I think we have now a clearer explanation as to why we use a learning experiment (see below; first paragraph of the result section):

“We taught participants (N = 210) to categorize a subset of our 80 objects in terms of their scores along a target dimension, and tested whether their learning could be generalized to a subset of untrained items. We considered an untrained categorization task. However, determining which dimension participants would use (or whether they would use one single dimension consistently throughout the experiment) would likely result in large differences in interpretation of the task across subjects. Instead, we used a learning paradigm to specifically test whether participants could reliably learn the organization of each dimension.”

Importantly, it is really not clear to us what would the alternative task be. Either we would give participants the label of the dimension – and we do not think this is what we wanted to do, given that then we would not be testing the actual dimensional values and the distribution of the objects by each dimension, but rather the understanding participants had of the labels collated from another set of participants; or we would not give any instruction and ask participants to categorize the objects freely. The latter possibility raises, however, and in our view, a lot more problems than the one we opted for. Specifically, we show that several dimensions can effectively structure the multidimensional space of each knowledge type. Thus, there could be large heterogeneity in the dimensions used per participant (and even within participant at different times of the experiment) when categorizing the objects presented. That would make it very hard for any conclusion to be derived.

Nevertheless, as the Reviewer suggests, this is a methodological problem that does not change the theoretical problem – that, in her/his own words, “Even if the categorization behavior of the third group of subjects is based on implicitly acquired knowledge, it is still based on acquired knowledge and not on spontaneous behavior”. The reviewer is right, obviously, that the fact that we use training is a limitation in terms of the spontaneity of these dimensions. Because of that, we have now added the following paragraph in the discussion (see below; antepenultimate paragraph of the discussion):

“Beyond the neural and content-specific results described above, a major aspect of our data relates to the ability of these dimensions to predict behavioral categorization judgements. Specifically, we trained participants to categorize objects while implicitly using a metric based on (one of) these dimensions. Critically, we showed that their learning could be generalized to untrained items (cf. the lack of generalization for our control dimensions). These data strongly suggest the validity of these dimensions in organizing the conceptual space of manipulable objects, as well as that of our approach to understanding the multidimensionality of object space. However, the use of a learning paradigm, rather than a less constrained paradigm where participants were free to use any dimension, is a limitation of our study, and may raise doubts as to the centrality and spontaneity of these dimensions to manipulable object processing. We believe, however, that in order to understand the role of each of these dimensions for object categorization and processing we needed to direct participants, implicitly, to our target dimensions, given the multitude of dimensions we obtained per knowledge type. Moreover, some of our data may help address this limitation. Specifically, we do not obtain generalization effects for the lexical frequency control dimension – a major linguistic dimension. Furthermore, we see a clear trend for stronger generalization effects for the dimensions that show higher stress values in the MDS solutions. These facts, and particularly the latter one, strongly suggests that specificity in the learning and generalization to novel objects is unique to the key object-related dimensions and not just to any dimension, and thus that these dimensions differentially structure how we think about manipulable objects. That is, transfer of the learning is not uniform but rather correlates with how central that dimension is for understanding how we represent objects in each knowledge type.”

Despite this limitation, we still think that our data allows us to derive our conclusions. As the Reviewer her/himself puts, the problem is that “In principle, all conceivable object properties can be described and trained dimensionally, without telling us anything about the naturally existing mental or neural representation of objects”. This is the crux of the issue – is it really the case that all conceivable properties/dimensions can be trained, and, more importantly, have that training be generalized to novel items? While we tend to agree with the first part – that almost anything can be learned – we think we did go to great length to show that the second part – i.e., the generalization part - is not necessarily true, and our data within our own key dimensions suggests this to be also incorrect.

We tested control dimensions to address this issue. Specifically, learning a lexical frequency-based dimension – a dimension that is central to the processing of linguistic stimuli, and that is highly prevalent

in cognitive processing – did not result in generalization. Perhaps more important to the point here, though, there was a clear trend of lower R^2 fits as a function of the lowering of the importance of the dimension for the MDS solution. All these dimensions refer to an “object property”, as described by the Reviewer, and again, as potentially proposed by the Reviewer, the ability to transfer their learning should be equal for all properties. Our data shows quite the contrary: transfer of the learning is not uniform but rather correlates with how central that dimension is for understanding how we represent objects in each knowledge type. That is, reduced generalization of learning for dimensions with lower values of stress is actually a demonstration of the specificity of this learning to object-related dimensions, and to the centrality of these dimensions – what we propose, and it seems logical to us, is that this centrality is mirroring the importance these dimensions have in structuring our reasoning about manipulable objects.

Finally, and despite the fact that we still believe our data allows for some conclusions about the centrality of these dimensions, we believe we have toned-down our conclusions on this. We hope the Reviewer agrees with us, and that the view of the Reviewer is correctly discussed.

1.2: The fact that different groups of participants were used does not change this conceptual restriction. The problem of reasoning is exacerbated by the fact that the entire construct is based on pre-specified categories. This decision is based on (certainly reasonable and plausible) presuppositions about how object knowledge is represented; but they remain presuppositions, so that all further conclusions must always be considered with this limitation. The rationale for the authors of this study choosing a training design is found somewhat late and rather incidentally in the statement “We used a training task [...] as there were several dimensions per knowledge type that could be used by the participants if we opted for an untrained experimental situation”. This reasoning is absolutely plausible, but unfortunately it also spoils the conclusion that can be drawn from the data obtained. These limitations should be clarified in the manuscript, and authors should avoid over-interpretation.

R: We hope the Reviewer agrees that 1) we have included and discussed this issue as a limitation in the manuscript; 2) we have included a clear reasoning for why we used this paradigm; 3) we have included a discussion of why we still think our data tells us more than what the Reviewer suggested, and that this reasoning is relatively sound (although the Reviewer may still not agree with it); and 4) we have toned-down some of our original conclusions. We nevertheless thank the Reviewer for raising this comment, as we believe that this issue was not discussed in the original manuscript, but should have been.

(2) Using these dimensions to model brain activity in a different set of subjects is a compelling approach. Since the task of these subjects was an object-animal categorization, the results regarding the effects of the dimensions appear to be valid. A criticism of the fMRI approach and analysis is that it would have been more plausible to use an RSA rather than a univariate approach via parametric modulators reflecting the dimensions used. Authors should explain why they did not choose the former approach.

R: We thank the Reviewer for the comment. Although we respectfully disagree with the contention that RSA would be a better method (see below), we have included in the revised manuscript a paragraph that discusses why we use parametric modulations as our analytical pipeline (last sentences of the first paragraph of the results):

“We used parametric analysis^{94–96}, casting our key dimensions as first-order (i.e., linear) parametric modulators in a General Linear Model (GLM; see Figure 1C), and asked whether the scores of each object in each dimension were able to explain neural responses elicited by those same objects. Using parametric modulations is the most appropriate approach because of the continuous nature of the scores of the objects in the dimensions extracted with MDS. Under this approach, we can directly test (i.e., without

transforming the data) whether the responses of a voxel are a function of the scores in the target dimension.”

We thought thoroughly about this issue. The reason why we do not agree that RSA would be the best method here – or, in fact, that parametric modulation is the best method for our data and purposes – is twofold:

a) Our dimensions are, in effect, 80X1 vectors, with each cell corresponding to the score of each object on that dimension. Moreover, these cell values are quasi-continuous, and report how much an object “has” of that dimension. For parametric analysis, all we need is to take this vector and include it as a modulator predictor in the GLM model – it is a direct and simple process. For RSA, we need an 80X80 RDM. This means that we will have to transform our vectors into an 80X80 matrix. This may seem straightforward, but, in fact, requires assumptions to be made. For instance, should the distance between two objects be the absolute value of the difference of dimensional scores? If we accept this assumption, then directionality information would be destroyed, if we do not accept it, then this creates a non-symmetrical matrix. What we mean to say is that these assumptions will have consequences. Overall, then, methodologically, parametric modulations seem the most straightforward analysis to be applied to our data.

b) Parametric modulation fits better with our purposes and goals. This is obviously the most important aspect of why we believe it to be the better method. What we want to see is whether the responses of each voxel change as a function of the dimensional scores. This is exactly what parametric modulations measure. RSA may measure this indirectly by looking at whether similarity at the dimensional level (i.e., distance between two objects) matches neural similarity. But it does not focus on whether variations in dimensional strength are captured by the voxels being analyzed.

We hope that the Reviewer sees the merit of our methodological decision.

(3) At first glance, the labels of the dimensions seem to help make neural representations interpretable. In fact, however, exactly this transfer is not - and I would also say: cannot be - provided by this study. For example, what exactly does it tell us that cooking objects activate the right pMTG more than sports objects do? One could speculate in what ways cooking and sports objects differ, and it is evident that they do so on many different (even more fine-grained) dimensions. But which of these many options explains the neuronal differences remains unaddressed. I would not say that it is not an interesting or valid finding that the right pMTG is more active for cooking objects, but it does not lead us to a better understanding of what exactly rules the (presumably multidimensional) neural representation of objects. The authors repeatedly emphasize that, in contrast to previous studies, they choose a finer within-category approach instead of distinguishing between (coarse) categories. In doing so, however, they seem to forget that the finer within-category dimensions were dependent on the coarser pre-selected categories. Thus, the question of object representation is possibly only shifted to another level, but not answered fundamentally differently: We see the same pattern of regional differences along dimensions as well as we can identify these differences at the categorical level. However, the nature of these differences - see the example of cooking vs. sports objects - is not clarified. Interestingly, in the final statement of their discussion, the authors arrive at a more realistic interpretation of their results. It would be important that speculative and over-interpretative parts of the discussion are reduced appropriately.

R: We thank the Reviewer for raising this concern. We agree with the fact that the labeling is a double-edged sword: it does allow us to describe the neural and behavioral results, but also is just a proxy to what the dimension might really be.

However, we think the revised manuscript addresses this issue, potentially indirectly, as we believe that what we emphasize, for the most part, is not really the specifics of the label assigned to each dimension, but rather the general understanding of that label. That is, we do not make claims about “metal” in the Metal vs. Other materials, but rather that that dimension is reflecting the material composition of the objects, especially in comparison with the more geometric dimensions (e.g., elongation, size). The same can be applied to the functional dimensions: we are not focusing on cooking or office, but rather on structuring objects as to whether they are present in particular contexts or whether they are referring to different action goals (e.g., cooking) – that is, context in which objects are seen vs the action goals we can fulfill with them. We also feel that the problem is perhaps stronger for function dimensions, because of their more abstract nature. Is this concern also important for elongation or grasp type? That we have voxels whose responses fluctuate as a function of the scores in these dimensions may not be as hard to “grasp” as for a dimension such as cooking vs sports, unless we conceive of this functional dimension as a proxy for types of action goals we achieve with these objects.

Thus, we do believe that our data gives us novel information as to how we structure manipulable object information, given the deep changes we made to the manuscript. Nevertheless, we clearly think we have answered the request put forth by the Reviewer. As the Reviewer puts it “It would be important that speculative and over-interpretative parts of the discussion are reduced appropriately.” We think we have clearly toned-down our conclusions, and most importantly, put it transparently where we are more speculative in our conclusions. We hope the Reviewer agrees!

(4) The discussion of the fMRI findings seems largely post-hoc and addresses a variety of brain regions. This gives the results something arbitrary. Even if several of the regions discussed later are mentioned in the Introduction, it is nevertheless clear that an exact assignment in the sense of a hypothesis testing did not take place and was probably not the most important goal of the study. Since the dimensions were derived from behavioral results in a first step, one could (but did not) have formulated neurofunctional hypotheses for these dimensions. I suggest that the post-hoc character should be explicitly pointed out. Also, parts of the discussion are starting already in the results section, something seems to have gotten mixed up there. Finally, the discussion is really too lengthy and often there are redundant paragraphs, so it can and should be shortened.

R: This is a major concern that was raised by all Reviewers. We completely agree that the way we wrote the original manuscript clearly read as post-hoc. We believe, however, that this was more of a problem of structuring the manuscript, rather than the actual lack of hypotheses. In fact, the Reviewer her/himself puts it that “the dimensions were derived from behavioral results in a first step”, and this made it hard to structure a manuscript to show our real hypotheses. Nevertheless, and because of this concern raised by the Reviewers, we managed to properly put out our predictions and separate these from the more speculative parts of the discussion. Both our introduction and conclusion were completely changed to adequately show what we predicted – at the level of the kinds of general dimensions we should get, as well as to the kinds of neural results we should obtain, and at the level of the potential expected and unexpected results, and how to speculatively explain unexpected results. Please see the last 5 of paragraphs of the introduction, and the second, third and fourth paragraphs of the discussion.

We believe these novel structure of the manuscript responds adequately to the concern of the Reviewer. Moreover, we have also streamlined the results section, as a response to the Reviewer’s comment. We truly thank the Reviewer, as her/his comments definitely improved dramatically our

manuscript. We hope the Reviewer agrees and sees sufficient merit for suggesting publication in Communications Biology.

Reviewer #2:

This paper is designed to uncover the nature of object knowledge, with specific focus on within category organization as many studies have mapped between category dimensions already. First, the authors acquired human subjective judgments on a large set of manipulable objects on several dimensions i.e., vision, function and manipulation which were explainable and generalizable. They then uncovered neural regions associated with the dimensions and conclude that the same dimensions govern the neural profile primarily in areas with manipulable-object preferences compared to other categories of objects.

The approach adopted is interesting but there are several concerns which undermine this reviewer's enthusiasm, some of which are methodological, some more theoretical.

R: We thank the Reviewer for her/his assessment of our manuscript and for all the work. We think that many of the concerns raised were of central importance for revising our manuscript. We hope that the Reviewer agrees that the new manuscript is greatly improved, and that it merits publication in *Communications Biology*

a. It is interesting that there was a major divide in the neural correlates of the two dimensions with material properties represented in more medial ventral temporal cortex, and in FG and LO. Are there a priori reasons why this should be? Suggesting that this might be related to mid-level properties is not a strong enough explanation unless you can demonstrate this is in some way otherwise this seems more a re-description of the findings.

R: We want to start by thanking this and the other Reviewers, as their comments on the post-hoc nature of our conclusions contributed to a major change in the organization of the manuscript. In fact, this comment (and the one below) along with similar comments from the other Reviewers made us realize that we did not put forth the hypotheses that we had about the content of the dimensions and the way in which these dimensions should explain neural responses elicited by the presentation of manipulable objects. Specifically, as already discussed in points 3 and 4 of Reviewer 1, the original manuscript was written indeed as if all our explanations of the neural data were post-hoc. That did not correspond to reality, as we did have predictions, but the structure of the original manuscript precluded us from putting them upfront. The structure of the revised manuscript clearly allows us to put forth all of our predictions (see the last 5 paragraphs of the introduction, as well as the second, third and fourth paragraphs of the discussion).

In what regards the concern expressed here by the Reviewer, we had clear predictions about the visual dimensions and their explanatory power. We have now included a paragraph in the introduction to discuss these predictions and the many established dissociations between different types of visual information (second to last paragraph of the introduction):

“Finally, we expect that the way in which we understand the visual properties of an object to be organized under two major types of object-related dimensions: those that relate to geometric properties (e.g., shape, size, elongation); and those that relate to surface and material properties (e.g., type of material, shininess, color). This distinction has been shown consistently in several single case patient studies. For instance, Patient DF presented with a clear deficit in processing visual form in the context of spared processing of surface and material properties^{81,82,83}, whereas patient MS presented with deficits that are specific to the processing of surface properties, in the context of spared shape information^{82,83}. This distinction is further supported by neuroimaging studies that demonstrate that more medial aspects of ventral temporal cortex (from lingual gyrus anteriorly to the parahippocampal gyrus) participate in the

computation of surface and material properties of the visual stimuli⁸²⁻⁸⁵, whereas more lateral and posterior aspects of ventral temporal cortex^{82-84,86-88}, and dorsal occipital cortex^{19,69,89-91} code for geometric properties. Thus, we expect the dimensions we obtain that structure visual similarity to relate, independently, to geometric and to surface properties and materials, and to conform to these lateral-to-medial, posterior-to-anterior neural dissociations.”

That is, both neuroimaging and neuropsychological data point to the clear-cut distinction and double dissociation between two (perhaps three if you consider size as independent from other geometric properties) major visual dimensions: those that code for geometric properties (e.g., elongation, tapering, size), and those that code for material and surface properties (e.g., different materials, smoothness). Moreover, this distinction has clear anatomical correlates – the lateral-to-medial gradient described.

We hope that this new formulation responds to the concerns of the Reviewer.

b. The analysis of the imaging data requires strengthening. The data are all analysed in terms of which regions are significant for which dimension (related to question in a). Seems that a better approach is to derive measures of the relationship between the dimensions and the neural signal. The analysis is by fiat – there is activation so this must be significant. Seems to be the wrong way around? More detailed examination and exploration is required.

R: We are not sure if we completely understand the Reviewer on this point. If we understood it correctly, though, we believe the response to this concern is two-fold: 1) The results of the parametric analysis are, in fact, directly what the Reviewer is asking for – that is, what we obtain is precisely a measure of the relationship between the change in dimensional scores and the change in neural signal per each voxel. We enter the object-specific scores of our dimensions (i.e., how much an object “has” of that dimension) in our GLM and see how much these scores explain the variation in neural signal elicited by the presentation of the 80 manipulable objects; and 2) it also seems like the Reviewer is alluding to the post-hoc nature of our original discussion (hence the analysis by fiat). We have thoroughly changed that, and have now a clear set of hypotheses – some of these are confirmed by our results, others are not, and yet other results were unpredicted. We then speculate about some of these issues, but we clearly state the speculative nature of some of our conclusions.

Overall, we hope that our answer to this concern fits with what the Reviewer was thinking when she/he put forth this concern.

c. The behavioral data are, in general, stronger than the neural data but I do not understand why there is training? If the knowledge base of observer is already configured along these dimensions (which it should be if this is the basis of object knowledge). The authors state that “as there were several dimensions per knowledge type that could be used by the participants if we opted for an untrained experimental situation”. But isn't the whole point to establish the dimensions? Much of the paper is presented as though this is the goal. Otherwise, the findings seem to show that when one trains participants, they learn? Okay, the dimensions do differ but this does not indicate the native knowledge structure of the participant.

R: We thank the Reviewer for this comment. We agree that this aspect of whether our behavioral task shows the centrality of our dimensions should have been discussed in the original manuscript. We have addressed this exact issue in Point 1 of Reviewer 1. There, and in the revised paper, we now discuss why we use a learning task, and suggest that the other possibilities (e.g., direct categorization of items) would

lead to methodological problems that, in our view, are far superior from the one exposed by the Reviewers (e.g., heterogeneity of the dimensions used, and thus difficulty in extracting any conclusions).

Moreover, we also discuss why we think that both the control dimensions and the results obtained do give strength to the conclusions we presented. Specifically, we show that not all “dimensions” allow for generalization of learning: firstly, the lexical frequency-based dimension (a major variable that we know affects strongly many cognitive processes) does not lead to transfer of learning to novel items. This is a starting point to show that the dimensions extracted are, in a way, “special”. Perhaps more importantly, we see decreases in the generalization to novel items as we move towards the dimensions with lower stress values on the MDS solution. That is, the importance of each dimension in structuring the way we think about these objects – measured by the stress value in the MDS solution – has an impact in how strong of a generalization to novel items we observed.

Overall, then, although we do use a training task, we do so just to implicitly prime participants towards the target dimensions, and the results on the generalization part of the experiment (i.e., the crucial part of the experiment) seems to track the importance of these dimensions in framing how we think about manipulable objects within each knowledge type.

d. Do you think there is activation in other parts of cortex aside from the areas included in Figure 4 composite dimension map. In its current form, it is misleading and would be better to have reader see the entire distribution of activation for each dimension. Is it indeed the case that when you plot F map against zero, there is no other activation? How would you reconcile this with the very many results that show dorsal cortex activation for affordance from manipulable objects? Or anterior temporal lobe for functional knowledge?

R: We thank the Reviewer for bring this up. In Supplementary Figure S4 (and in Figure 5 for the first two dimensions of each knowledge type) we present the corrected maps per dimension independently. Those are whole brain maps that are corrected for multiple comparisons. So, if the Reviewer’s question is whether we are masking any results from the maps that survive correction, the answer is no. Those are the full, whole brain maps that survive correction for multiple comparisons per dimension. Thus, those are the voxels whose variation in neural response is explained significantly by variation in dimensional scores.

Below, we present $p < 0,05$ uncorrected maps F maps per dimension – we can add this to the supplementary materials if the reviewer thinks it is beneficial, although we are not sure whether that will improve the interpretability. In fact, we believe that that could be misleading as nothing can be derived from uncorrected maps due to the multiple comparisons performed.

As can be seen below, these maps show a lot more coverage, and importantly in predictable and interesting locations. For instance, manipulation dimensions seem to be capable of explaining signal within many more parietal and motor regions. Moreover, the dissociations between visual dimensions (between geometric, size and material dimensions) become even more clear when we look at these maps.

Functional Dimension 1

Functional Dimension 2

Functional Dimension 3

Functional Dimension 4

Manipulation Dimension 1

Manipulation Dimension 2

Manipulation Dimension 3

Manipulation Dimension 4

Manipulation Dimension 5

Manipulation Dimension 6

Vision Dimension 1

Vision Dimension 2

Vision Dimension 3

Vision Dimension 4

Vision Dimension 5

Regardless of the inclusion of these uncorrected maps on the manuscript, the Reviewer raises a very important question – how do we square our results (or the lack thereof) with the literature. The revised manuscript does a much better job at resolving this issue, as we now have predictions and then discuss the results in light of these predictions.

Below I present our predictions for the manipulation dimensions (antepenultimate paragraph of the introduction):

“Dimensions that structure our understanding of the manner in which an object is manipulated should relate both to motor aspects that are directly available from the visual input (e.g., object affordances such as grasp types; e.g.,⁵⁹⁻⁶⁷), as well as to aspects that may have to be derived from the visual input and are part of an object-specific manipulation program (e.g., object-related specific movements such as different wrist rotations). These dimensions should be able to explain responses in regions that relate to the processing of affordances and praxis. These include occipito-parietal and posterior and superior parietal cortical regions (in the vicinity of V3A, V7, and IPS) that have been shown to be important for the computation of object grasps and object affordances⁵⁹⁻⁶⁷, and for 3D object processing⁶⁸⁻⁷⁸. Moreover, the left IPL, which has been shown to be causally involved in processing object-specific praxis^{14,51,79,80}, should also be explained by dimensions structuring manipulation similarity.”

Thus, we do predict that manipulation dimensions should explain dorsal occipital, and posterior/superior and inferior parietal cortical activations – as these predictions follow the results obtained in the literature. This would be especially true for those dimensions that are more affordance-based. In our view, the main affordance-based dimension is the first dimension – i.e., the “power vs. precision” dimension. The information reflected in the other dimensions has to be derived after some sort of recognition - how does one know that a bottle cap needs to be rotated if not after recognizing that it is

a bottle cap (compare this to realizing what grasp to use when interacting with an object, an aspect that can be directly inferred from the visual stimulus without object identification, and thus truly a motor affordance of the kind highly related with processing within the dorsal stream).

Importantly, and as described in the discussion, we partially corroborate those hypotheses. See below the paragraph added to the discussion on the results from the manipulation dimensions (third paragraph of the discussion):

“The content covered by manipulation dimensions also conforms, in part, to what was predicted, in that we obtained dimensions that relate strongly to object affordances – in particular, the type of grasp afforded by the manipulable objects used – and to motor aspects that are not necessarily directly available from the inspection of an object (e.g., “Rotation”). Thus, posterior parietal cortex and dorsal occipital cortex, including V3A, IPS (extending to post-central gyrus), and Precuneus – areas involved in hand shaping for grasp, grasp planning, processing different types of grip formations^{32,44,59,60,62,64,90,104–108} (but see^{60,61}), and coding for shape properties and 3D representations of objects independently of action^{70,72–75,82,89,109} – were predictably explained by manipulation dimensions, and especially the “Power vs. Precision” dimension. Notably, several aspects regarding the manipulation dimensions were not in line with our initial expectations. Firstly, one of the major dimensions of manipulation – “Dexterity vs. Force” – may be conceived more of as a dimension that is important for interacting with an object that is less motor-related than the other dimensions obtained. In fact, and perhaps speculatively, the involvement of ventral regions under this dimension (both posteriorly within lingual gyrus, and anteriorly within parahippocampal gyrus), is consistent with the importance of surface and material properties in deriving the weight of an object¹¹⁰, and thus the amount of force or dexterity needed to act with these objects¹¹¹. Secondly, we predicted manipulation dimensions would explain responses in more anterior areas of parietal cortex, and namely the left IPL, as this region is causally involved in the retrieval of praxis^{14,44,45,51,79,80}. Nevertheless, none of the manipulation dimensions obtained was able to explain responses within left IPL. This unexpected result may reflect the fact that the dimensions obtained relate to piecemeal aspects of the full-blown motor program associated with an object (e.g., grasp, wrist rotation), whereas left IPL seems to be engaged when retrieving an object’s unique full motor program^{51,80}.”

As it is clear in this revised version, we try to explain why some of the dimensions do not explain neural activation in certain areas – namely the lack of explanatory power within IPL for all manipulation dimensions, and within dorsal regions for the second dimension. We do so in the context of the state-of-the-art. Note, however, that some of these explanations are more speculative – and we now clearly state that. Moreover, note that in the uncorrected maps we can see activations in IPL and in motor cortex. Furthermore, in the uncorrected maps we do see anterior temporal cortical signal being explained by function dimensions.

Overall, we hope that the Reviewer agrees that the manuscript is now much clearer and stronger, and that we adequately responded to the suggestions and concerns raised. Thank you again for your work.

Reviewer #3:

Disclaimer: I was one of the reviewers at a previous journal. I appreciate the authors addressing many of the issues mentioned in the previous round of reviews, specifically regarding the framing of the work. I still see several outstanding issues that I think would be important to address.

R: We want to thank the Reviewer for her/his work in the two previous versions of this manuscript. I think it is clear that the comments and suggestions we have received have been central for the revision of the manuscript, and we thank the Reviewer for her/his contribution to this process. We truly hope that this revised manuscript is now much better at responding to the full set of concerns.

1. The main motivation of the authors has become a lot clearer through the revision, and the framing with a focus on manipulability (i.e. within-category) is much more in line with the results. I would appreciate if the authors were a bit more consistent with this framing, since many of the original formulations continue to be present in the manuscript that do not directly appear to follow from the results, specifically in the abstract but also in the main text (e.g. "multidimensionality is a signature of high-level object processing" or "perhaps in the same way as the different dimensions that rule the organization of low-level sensory-motor cortices overlap spatially, so do dimensions that rule the organization of object knowledge in the brain."). I understand that they aim at drawing more general conclusions but this alternative framing either appears too early in the manuscript without justification, or could be better justified later.

R: We have tried to go through the manuscript and consistently apply this framing. We hope the Reviewer agrees that we are now a lot more consistent!

2. Currently, it is also not entirely clear whether the focus of the work is on multidimensionality as such, which is then confirmed by the results in the brain, whether the focus is on identifying multidimensionality in the brain, for which the behavioral results are just a necessary element, or whether it is both (behavioral and neural representations). It would be good if the authors could frame this more consistently.

R: In our view, it has to be both the neural and behavioral aspects as what we want to show is that multidimensionality is the hallmark of manipulable object processing. To show that, we feel we need to demonstrate the role of dimensionality in the different levels in which we can probe the mind – its behavioral responses but also its neural responses. If this is not clear in the revise manuscript, we can still change the introduction to better reflect this.

3. For the interpretability task, one issue that I currently do not see addressed is whether participants knew which knowledge type they were assessed on (visual, function, manipulation). It just appears to be striking that participants could tell the knowledge types from each other so clearly, and it would be worth adding an analysis of this to the interpretability results if this truly was the case. It is, of course, fine if they were instructed of the knowledge type, even though this may somewhat limit the interpretability.

R: We are sorry for the lack of information on this regard. We did instruct participants on the target knowledge type. This is now clearly written in the methods section under the subheading "Label generation for extracted dimensions".

4. More generally, it would be very instructive to see how strongly the different dimensions are related with each other between knowledge types to better understand their relationship. If they are strongly correlated, they may be drawing on similar information. If they are weakly correlated, this would indicate a clear separation of information between knowledge types.

R: Thank you very much for drawing our attention to this issue. We have now calculated the correlation between the key dimensions (see below).

As can be seen, there are some correlations between the dimensions. Most of them small to moderate. There are, however, 8 correlations (out of the possible 77) that are above .5, all but one below .7 (maximum is - .71: between “Metal vs. Other materials” and “Kitchen vs. Office”).

Overall, then, correlations are rather low, with few exceptions. These exceptions may be expected – many of the kitchen utensils shown do have a metal part – and are part of the natural distribution of object properties in the world.

5. One additional reason why this may be relevant is the fact that the current brain-based analyses are executed separately for different domains. If these behavioral dimensions are strongly correlated between domains, then it might be good if the authors discussed this as a potential limitation of the interpretability of their results.

R: We separated GLMs into knowledge types because the dimensions from a knowledge type are orthogonal and thus, uncorrelated. As such, the separation between the maps is clear. As discussed above, however, there are a few correlations that can be considered strong. As described above, also, these correlations are probably a reflection of real-world natural correlations. Note, however, that there is also a degree of independence between even the most correlated dimensions. If the Reviewer believes this to be of sufficient importance to be in the manuscript, we will gladly include it.

6. In the learning task, is it possible participants solved the task merely by asking which object is more similar to the reference object with respect to function, manipulation, or vision? Since participants had been instructed, this alternative approach may, in fact, be quite simple yet would not indicate

interpretability of dimensions but merely the ability to report similarity to two reference objects. Perhaps the authors could run an analysis highlighting what the best possible performance would look like under this scenario.

R: Thank you very much for this comment. Let us start by saying that in very general terms, the set of dimensions structuring each knowledge type will necessarily reflect (some of) the overall similarity between objects as measured in our similarity judgement task. Nevertheless, the preliminary analysis we performed suggests that our results are really dependent on the dimensions more so than on the rank order of objects by similarity to the anchors in the general knowledge type specific (vision, manipulation, function) similarity matrix.

As proposed by the Reviewer, we tried to come up with an analysis to respond to this concern, but it turned out to be impossible to come up with a formal analysis for two reasons:

- a) The way in which the experiment was created – the divisions by bins, and the division of items within each bin as novel and learned – made it very hard to test whether general similarity could explain the results. This is because out of the 8 items in each bin, only three were “novel”, and thus, disentangling the overlap between these bin items and the anchor-specific rank-ordered general similarity items led to the presence of missing values.
- b) There were also many items that were very similar to both anchors but were on one of the extremes of the target dimension. The question is, how would one label those items in terms of being close to an anchor (i.e., dimension-wise they are close to one of the anchors but overall-similarity-wise they are close to both anchors)? By the way, by itself, this fact already suggests that the alternative advanced by the Reviewer may not actually be the culprit of our behavioral effects.

Nevertheless, and for the sake of our response to the Reviewer, we have some data to present to the that although we do not think could be in the paper (because it is potentially not statistically sound), responds adequately to the concern of the Reviewer.

As a first level of response, we looked at the overlap between the seventh closest items to each anchor and the items comprising the most extreme bins per dimension. Note that we did not take into account the order but only the membership to the bin/the 7 closest items. The overlap is of about 42%. If we were to look at actual rank order, then the overlap would be almost inexistant. We also looked at the second and second to last bins, and at the overlap between items in these bins and items within the subsequent 8 items in overall similarity with the anchors. Here, the similarity drops to 18%.

We then looked at the behavioral responses to novel items in the most extreme bins of the first dimension of each knowledge type. To test for the hypothesis advanced by the Reviewer, we looked exclusively at items that could disentangle the role of general similarity to the anchors and the role of dimensionality. That is, we only looked at the responses to items that were on the extreme bins for each dimension, but *were not* within the seventh most similar items to the anchors used on that particular dimension. We had to average across all participants (i.e., our analysis is a fixed effects analysis) because of the coverage issue describe above. These are the results:

	Function Dimension 1 (original)	Function Dimension 1 without anchor similarity	Vision Dimension 1 (original)	Vision Dimension 1 without anchor similarity	Manipulation Dimension 1 (original)	Manipulation Dimension 1 without anchor similarity
Bin 1	16%	8%	27%	38%	13%	8%
Bin 10	93%	100%	84%	94%	88%	86%

That is, for the most part, taking out the items that were very close in overall similarity to the anchors from the “novel item” group did not change the percent response performance towards one of the anchors when comparing it to using all of the novel items (if anything, it brings more extreme values...). Thus, we believe that this shows that our results are really specific to the dimensions extracted and not directly to the similarity item pool around the anchors used.

We hope this responds to the concern of the Reviewer.

7. Evaluating the overlap between clusters in a cluster analysis requires a conjunction analysis, specifically since a cluster surviving a cluster threshold can only be interpreted as "at least one voxel within the cluster shows information" (see Woo et al., 2014, Neuroimage, and Rosenblatt et al., 2018, Neuroimage). Without this, the inferences drawn about spatial overlap of activations are statistically invalid.

R: We thank the Reviewer for bring up this issue. We realized that in the original manuscript we mentioned that there was an overlap between the different dimensions. However, we did not mean statistically significant overlap, but rather just that different dimensions are able to explain activation IN OR AROUND similar areas – that is, we meant more proximity of explanatory power! We have changed the manuscript accordingly.

8. The brain results could be made clearer, and there still seems to be a lot of reverse inference at play. For example, “many of these dimensions overlap spatially in particular regions” is a very vague statement. “most probably due to the fact that these dorsal occipital regions are involved in object-specific 3D processing” appears like a strong reverse inference. The entire section “Object related-dimensions maintain content specificity in their neural responses” appears to be trying to fit the results within a framework of existing findings, which leads to a lot of reverse inference. Given the diverse findings associated with object processing regions, it may be better if the authors focused more strongly on their key message: that multidimensionality explains findings and that it may also appear to explain other aspects of what we know about the functional organization in these regions. Thus, rather than relating the dimensions to previous findings, why not relate previous findings to the dimensions, thus highlighting their explanatory value, and thus turning a reverse inference into more of a forward inference? What are key findings in relation to manipulability of objects and how do these results capture these findings (or do not capture them)

R: We thank the Reviewer for this comment. We followed this suggestion, and has already described in our responses to the other Reviewers, this was transforming – the manuscript was thoroughly changed, and we do think that it now reflects more correctly our thinking and our initial predictions. We hope the Reviewer agrees with us on this one!

Other comments:

9. Given that the cutoffs chosen by the authors in Figure 3 continue to be arbitrary, are not justified by the authors, and do not seem to serve a clear purpose, why not remove them?

R: We have removed them. Thank you! Moreover, we also noted and corrected a small mistake in the violin plots of the R^2 values (y-axis). Note, however, that the results are virtually equal.

10. There are still numerous typographic errors throughout the manuscripts (e.g. fine-grain instead of fine-grained). I think it would help if the authors had a thorough read through the manuscript to eliminate these.

R: We went through the manuscript, and we hope we caught most of the typos mentioned.

REVIEWERS' COMMENTS:

Reviewer #3 (Remarks to the Author):

I would like to thank the authors for addressing my comments and for trying to come up with analyses for overcoming some of my concerns. For some comments, the authors wrote that they would be happy to include a discussion of some of the topics I raised should I find it necessary. I think the paper would benefit from a critical discussion of the potential shortcomings of certain analyses but I would also be fine if the authors kept the discussion as it is.